# NTKMTL: Mitigating Task Imbalance in Multi-Task Learning from Neural Tangent Kernel Perspective

**Xiaohan Qin**[1,2], **Xiaoxing Wang**[1], **Ning Liao**[1], **Junchi Yan**[1,2,‡]

[1]School of Artificial Intelligence & School of Computer Science, Shanghai Jiao Tong University
[2]Shanghai Innovation Institute

## Abstract

Multi-Task Learning (MTL) enables a single model to learn multiple tasks simultaneously, leveraging knowledge transfer among tasks for enhanced generalization, and has been widely applied across various domains. However, task imbalance remains a major challenge in MTL. Although balancing the convergence speeds of different tasks is an effective approach to address this issue, it is highly challenging to accurately characterize the training dynamics and convergence speeds of multiple tasks within the complex MTL system. To this end, we attempt to analyze the training dynamics in MTL by leveraging Neural Tangent Kernel (NTK) theory and propose a new MTL method, NTKMTL. Specifically, we introduce an extended NTK matrix for MTL and adopt spectral analysis to balance the convergence speeds of multiple tasks, thereby mitigating task imbalance. Based on the approximation via shared representation, we further propose NTKMTL-SR, achieving training efficiency while maintaining competitive performance. Extensive experiments demonstrate that our methods achieve state-of-the-art performance across a wide range of benchmarks, including both multi-task supervised learning and multi-task reinforcement learning. Source code is available at https://github.com/jianke0604/NTKMTL.

## 1  Introduction

Multi-task learning (MTL) [9, 20, 56, 45, 61] involves training a single model to address multiple tasks concurrently. This approach enables sharing information and representations across tasks, enhancing the model's generalization capabilities and boosting performance on individual tasks [6, 50, 38]. MTL is particularly advantageous in scenarios with limited computational resources, as it eliminates the need to maintain separate models for each task. Its utility spans a wide range of domains, including computer vision [1, 63, 35], natural language processing [10, 34, 40], and robotics [16, 55]. Despite these benefits, MTL encounters a significant challenge known as task imbalance, where certain tasks dominate the training process while others suffer from insufficient optimization. Previous studies [4, 48] have indicated that achieving more balanced optimization across tasks often leads to improved overall performance. Addressing such a task imbalance issue necessitates the development of sophisticated optimization strategies to ensure that all tasks benefit equitably from the shared model parameters.

One widely adopted perspective to address the above issue is to balance the convergence speeds of different tasks [32, 60, 35]. Nevertheless, it is highly challenging to accurately analyze the training dynamics and convergence speeds of multiple tasks within the complex MTL system. Most prior methods [35, 31] approximate the convergence speeds based on the difference or ratio between consecutive loss values. However, different tasks exhibit vastly different loss scales and heterogeneous

---

‡ Corresponding author. This work is partly supported by NSFC (92370201).

ultimate loss minima, thus such a simple approximation fails to accurately capture a task's convergence capability or potential at a specific training stage. As demonstrated by experimental results on widely used benchmarks such as NYUv2, these methods still exhibit considerable task imbalance. Therefore, there is a pressing need for a tool to characterize MTL training dynamics and task convergence properties with a robust theoretical foundation.

To this end, we attempt to analyze the training dynamics within MTL systems by leveraging *Neural Tangent Kernel* (NTK) theory[17, 2, 7], which provides insights into the optimization trajectory of deep neural networks and has demonstrated its theoretical efficacy in single-task learning (STL) scenarios. From the perspective of NTK theory, the convergence speed of a neural network can be characterized by the eigenvalues of its corresponding NTK matrix. Specifically, lower-frequency components of the target function typically correspond to larger NTK eigenvalues, which converge faster [7, 52]. In contrast, higher-frequency components often correspond to smaller NTK eigenvalues, which converge more slowly (or are harder to learn). This phenomenon is known as "*spectral bias*" in the context of single-task learning [22, 43, 53], which bears a strong resemblance to task imbalance in MTL. As explained by the NTK theory, such a distinction in the training dynamics provides a foundational understanding of why certain aspects of the target function are prioritized over others during the training process. However, despite its potential relevance, the application of NTK theory to the field of MTL has been scarcely explored by prior work.

Building upon the above motivation, this work applies the NTK theory to MTL scenarios and introduces an extended NTK matrix to jointly characterize the training dynamics of multiple tasks. Specifically, the target function in MTL is explicitly decomposed into multiple distinct components, each associated with a different task. Our theoretical analysis shows that, the convergence speed of the overall training error is jointly influenced by the NTK matrices corresponding to each task, and tasks associated with larger NTK eigenvalues can be learned more rapidly by the network, subsequently dominating other tasks and resulting in unsatisfactory performance on the remaining tasks. To address this issue, we propose a new MTL approach, NTKMTL, which assigns appropriate weights during training based on the NTK analysis of each task. This method effectively balances the convergence speeds of different tasks and, consequently, reduces task imbalance. In summary, our contributions can be outlined as follows:

**1) We introduce a new perspective on understanding task imbalance in MTL by leveraging NTK theory for analysis.** Under this perspective, multiple tasks are viewed as distinct components of the training objective, each characterized by unique NTK spectral properties. These inherent differences in spectral characteristics lead to significant disparities in convergence speeds across tasks, subsequently causing task imbalance.

**2) Based on the NTK spectral analysis of different tasks in MTL, we propose a new MTL method, NTKMTL.** Both theoretical analysis and experimental results demonstrate that NTKMTL effectively addresses the ill-conditioned distribution of NTK eigenvalues during MTL training, thereby alleviating task imbalance.

**3) To enhance the practical applicability and computational efficiency, we further introduce NTKMTL-SR, an efficient approximation that leverages the _Shared Representation in MTL.** NTKMTL-SR requires only a single gradient backpropagation per iteration for shared parameters, which not only provides computational efficiency but also maintains competitive performance.

**4) Extensive experiments validate that our methods achieve state-of-the-art performance across a wide range of benchmarks.** The experimental scenarios include both multi-task supervised learning and multi-task reinforcement learning, with task numbers ranging from 2 to 40.

## 2   Related Work

**Multi-Task Learning.** In multi-task learning (MTL), previous approaches can be broadly classified into two categories: loss-oriented and gradient-oriented methods. Loss-oriented methods primarily aim to address convergence discrepancies arising from differences in loss scales, thereby mitigating task imbalance. These methods often exhibit training efficiency as they only require one backpropagation on the aggregated loss. Approaches include Linear Scalarization (LS), Scale-Invariant (SI), homoscedastic uncertainty weighting [24], dynamic weight averaging [35], self-paced learning [37], geometric loss [14], random loss weighting [29], impartial loss weighting [33], fast

adaptive optimization [31], and multi-task grouping for alignment [48]. On the other hand, gradient-oriented methods [11] focus on resolving the gradient conflicts issue among shared parameters, seeking the most favorable updating vector to alleviate task imbalance. These types of methods often achieve better performance due to directly obtaining the gradients for all tasks for optimization. Notable approaches in this category include Multiple Gradient Descent Algorithm [46], gradient normalization [12], gradient conflicts projection [57], gradient sign dropout [13], impartial gradient weighting [33], conflict-averse gradients [32], gradient similarity regularisation [51], dual balancing [27], stochastic direction-oriented update [54], smooth tchebycheff scalarization [30], independent gradient alignment [47], Nash bargaining solution [39], fair resource allocation [4], performance informed variance reduction approach [41], and consistent multi-task learning with task-specific parameters [42].

**Neural Tangent Kernel theory.** Recent theoretical advancements have modeled neural networks in the limits of infinite width and infinitesimal learning rate as kernel regression using the Neural Tangent Kernel (NTK) [17, 2, 7, 22]. Specifically, analyses by [26] and [2] demonstrate that, during gradient descent, the outputs of a neural network remain close to those of a linear dynamical system, with the convergence speed governed by the eigenvalues of the NTK matrix [5, 7, 52, 53]. The NTK's eigendecomposition reveals that its eigenvalue spectrum decays rapidly as a function of frequency, which explains the well-documented "spectral bias" of deep networks towards learning low-frequency functions [22, 43, 53]. Such spectral bias phenomenon bears a strong resemblance to task imbalance in MTL, which is an effective application of NTK theory in explaining neural network training dynamics in STL scenarios. Building on this observation, this paper extends NTK theory to MTL and proposes a solution to address the task imbalance issue.

## 3 Method

### 3.1 Preliminaries

To establish the foundation for our theoretical investigation, we first review the traditional Neural Tangent Kernel theory that explores the training dynamics of deep neural networks. In the subsequent sections, we extend these theoretical frameworks to the multi-task learning context and develop specific methodologies accordingly.

**Neural Tangent Kernel.** For a deep neural network $f$ with parameters $\theta$ and training dataset $(\mathbf{x}, \mathbf{y}) = \{(x_i, y_i)\}_{i=1}^n$, the Neural Tangent Kernel (NTK) $\mathcal{K}$ is defined as

$$\mathcal{K}_{uv} = \left\langle \frac{\partial f(\theta, x_u)}{\partial \theta}, \frac{\partial f(\theta, x_v)}{\partial \theta} \right\rangle. \tag{1}$$

Prior works [22, 26, 2] have demonstrated that under certain conditions (e.g., when the learning rate $\eta$ approaches zero), the training dynamics of the neural network can be characterized via gradient flow, which is governed by the following ordinary differential equation (ODE) [44]:

$$\frac{d\theta(t)}{dt} = -\nabla \mathcal{L}(\theta). \tag{2}$$

This leads to the following theorem:

**Theorem 3.1.** *Let $\mathcal{O}(t) = \{f(\theta, x_i)\}_{i=1}^n$ be the outputs of the neural network at time $t$. $\mathbf{x} = \{x_i\}_{i=1}^n$ is the input data, and $\mathbf{y} = \{y_i\}_{i=1}^n$ is the corresponding label, Then $\mathcal{O}(t)$ follows this evolution:*

$$\frac{d\mathcal{O}(t)}{dt} = -\mathcal{K} \cdot (\mathcal{O}(t) - \mathbf{y}). \tag{3}$$

Detailed analysis can be found in the Appendix. Eq. 3 enables us to utilize the neural tangent kernel to analyze the training dynamics of neural networks. Prior work [22] demonstrated that as the network width approaches infinity, the NTK $\mathcal{K}$ remains approximately constant during training. A more widely used result is that during the training of deep neural networks, the kernel function is updated much more slowly than the network's output [52, 62]. Therefore, Eq. 3 can also be interpreted as an ODE and provides the following approximation:

$$\mathcal{O}(t) \approx (\mathbf{I} - e^{-\eta \mathcal{K} t}) \mathbf{y}. \tag{4}$$

**Spectral analysis in neural network training.** Let us consider the training error $\mathcal{O}(t) - \mathbf{y}$. Since the NTK matrix must be positive semi-definite, we can take its spectral decomposition $\mathcal{K} = Q\Lambda Q^\top$, where $Q$ is an orthogonal matrix and $\Lambda$ is a diagonal matrix whose entries are the eigenvalues $\lambda$ of $\mathcal{K}$. Then, since $e^{-\eta\mathcal{K}t} = Qe^{-\eta\Lambda t}Q^\top$, we have

$$Q^\top(\mathcal{O}(t) - \mathbf{y}) \approx -Q^\top e^{-\eta\mathcal{K}t}\mathbf{y} = -e^{-\eta\Lambda t}Q^\top\mathbf{y}. \tag{5}$$

This implies that, when considering the convergence of training in the NTK eigenbasis, the $i$-th component of the absolute error $\left|Q^\top(\mathcal{O}(t) - y)\right|_i$ decays at an approximate exponential rate of $\eta\lambda_i$. In other words, the components of the target function corresponding to kernel eigenvectors with larger eigenvalues are learned more rapidly, resulting in the high-frequency components of the target function converging exceedingly slowly, to the extent that the neural network is difficult to learn these components.

## 3.2 Extended NTK in Multi-Task Learning

In multi-task learning, a neural network with shared parameters $\theta$ is trained to simultaneously learn $k$ distinct tasks. In the general case, the overall loss function is defined as

$$\mathcal{L}(\theta) = \sum_{i=1}^{k} \ell_i(\theta). \tag{6}$$

In this context, by leveraging the gradient flow defined in Eq. 2, Theorem 3.1 can be further extended as follows:

**Theorem 3.2.** *Let $\{\mathcal{O}_1(t), \mathcal{O}_2(t), \ldots, \mathcal{O}_k(t)\}$ denote the outputs of the neural network function $\{f_1, f_2, \ldots, f_k\}$ for the $k$ tasks at time $t$, and let $\{\mathbf{y}_1, \mathbf{y}_2, \ldots, \mathbf{y}_k\}$ represent the corresponding labels. Then, the ordinary differential equation in Eq. 2 gives the following evolution:*

$$\begin{bmatrix} \frac{d\mathcal{O}_1(t)}{dt} \\ \vdots \\ \frac{d\mathcal{O}_k(t)}{dt} \end{bmatrix} = -\underbrace{\begin{bmatrix} \mathcal{K}_{11} & \cdots & \mathcal{K}_{1k} \\ \vdots & \ddots & \vdots \\ \mathcal{K}_{k1} & \cdots & \mathcal{K}_{kk} \end{bmatrix}}_{\widetilde{\mathcal{K}}} \begin{bmatrix} \mathcal{O}_1(t) - \mathbf{y}_1 \\ \vdots \\ \mathcal{O}_k(t) - \mathbf{y}_k \end{bmatrix}, \tag{7}$$

*where $\mathcal{K}_{ij} \in \mathbb{R}^{n \times n}$ and $\mathcal{K}_{ij} = \mathcal{K}_{ji}^\top$ for $1 \le i, j \le k$. The $(u, v)$-th entry of $\mathcal{K}_{ij}$ is defined as*

$$(\mathcal{K}_{ij})_{uv} = \left\langle \frac{\partial f_i(\theta, x_u)}{\partial\theta}, \frac{\partial f_j(\theta, x_v)}{\partial\theta} \right\rangle. \tag{8}$$

Detailed proof can be found in the Appendix. We define the large NTK matrix, formed by the training dynamics of the $k$ tasks in Theorem 3.2, as the extended NTK matrix in MTL, denoted as $\widetilde{\mathcal{K}}$. It is straightforward to observe that $\mathcal{K}_{ii}$ for $1 \le i \le k$, as well as $\widetilde{\mathcal{K}}$ itself, are positive semi-definite matrices. In fact, let $J_i$ denote the Jacobian matrix of $f_i$ with respect to $\theta$, then we can observe that:

$$\mathcal{K}_{ii} = J_i J_i^\top, \text{ and } \widetilde{\mathcal{K}} = \begin{bmatrix} J_1 \\ \vdots \\ J_k \end{bmatrix} \begin{bmatrix} J_1^\top & \cdots & J_k^\top \end{bmatrix}. \tag{9}$$

Analogous to the analysis presented in Sec. 3.1, we adopt the approximation derived from the theoretical framework of [22]. Consequently, Eq. 4 is extended to the multi-task learning scenarios as follows:

$$\begin{bmatrix} \mathcal{O}_1(t) \\ \vdots \\ \mathcal{O}_k(t) \end{bmatrix} \approx (\mathbf{I} - e^{-\eta\widetilde{\mathcal{K}}t}) \begin{bmatrix} \mathbf{y}_1 \\ \vdots \\ \mathbf{y}_k \end{bmatrix}. \tag{10}$$

As mentioned above, the extended NTK matrix is positive semi-definite. Therefore, similarly to previous steps, we perform its spectral decomposition as $\widetilde{\mathcal{K}} = \widetilde{Q}\widetilde{\Lambda}\widetilde{Q}^\top$. Consequently, the training error MTL can be approximated by the following evolution:

$$\widetilde{Q}^\top\left(\begin{bmatrix} \mathcal{O}_1(t) \\ \vdots \\ \mathcal{O}_k(t) \end{bmatrix} - \begin{bmatrix} \mathbf{y}_1 \\ \vdots \\ \mathbf{y}_k \end{bmatrix}\right) \approx -e^{-\eta\widetilde{\Lambda}t}\widetilde{Q}^\top\begin{bmatrix} \mathbf{y}_1 \\ \vdots \\ \mathbf{y}_k \end{bmatrix}. \tag{11}$$

Eq. 11 illustrates that in the MTL scenario, where the learning objectives are explicitly partitioned into $k$ components, the training dynamics can still be interpreted through spectral analysis of the extended NTK matrix. The substantial disparities in the distribution of NTK eigenvalues across different tasks give rise to bias in convergence speed, causing the network to be dominated by certain specific tasks and hindering the effective simultaneous learning of all tasks.

### 3.3 The Proposed NTKMTL

Theoretical analysis in Sec. 3.2 and experimental results on extensive benchmarks indicate that the traditional multi-task optimization objective in Eq. 6 yields unsatisfactory performance, exhibiting significant task imbalance across various task scenarios. Therefore, existing MTL methods often consider obtaining a weight $\boldsymbol{\omega} = (\omega_1, \omega_2, \ldots, \omega_k)^\top$ to optimize the weighted objective instead:

$$\mathcal{L}(\theta) = \sum_{i=1}^{k} \omega_i \ell_i(\theta). \tag{12}$$

Specifically, the final gradient is $g_s = \sum_{i=1}^{k} \omega_i g_i$, where $g_i$ represents the gradient of the $i$-th task's loss $\ell_i$ with respect to the shared parameters, and $g_s$ is the final aggregated gradient. Under this formulation, the analysis of the extended NTK matrix for MTL systems presented in Sec. 3.2 is correspondingly modified. Therefore, we further introduce the following proposition:

**Proposition 3.3.** *(Extension of Theorem 3.2) Let $\{\mathcal{O}_i(t)\}_{i=1}^{k}$, $\{\mathbf{y}_i\}_{i=1}^{k}$, and $\{\mathcal{K}_{ij}\}_{1 \leq i,j \leq k}$ be defined as in Theorem 3.2. We now replace the MTL optimization objective with the weighted form as presented in Eq. 12. Consequently, the ordinary differential equation governing the MTL training dynamics in Eq. 7 becomes:*

$$\begin{bmatrix} \frac{d\mathcal{O}_1(t)}{dt} \\ \vdots \\ \frac{d\mathcal{O}_k(t)}{dt} \end{bmatrix} = -\underbrace{\begin{bmatrix} \omega_1^2 \mathcal{K}_{11} & \cdots & \omega_1 \omega_k \mathcal{K}_{1k} \\ \vdots & \ddots & \vdots \\ \omega_k \omega_1 \mathcal{K}_{k1} & \cdots & \omega_k^2 \mathcal{K}_{kk} \end{bmatrix}}_{\boldsymbol{\omega}\boldsymbol{\omega}^\top \odot \widetilde{\mathcal{K}}} \begin{bmatrix} \mathcal{O}_1(t) - \mathbf{y}_1 \\ \vdots \\ \mathcal{O}_k(t) - \mathbf{y}_k \end{bmatrix}. \tag{13}$$

Detailed proof can be found in the Appendix. The NTK matrix derived from the weighted optimization objective in Eq. 12 remains positive semi-definite, and thus, the analysis in Sec. 3.2 is still applicable. Proposition 3.3 shows that the eigenvalues of the new NTK matrix are strongly correlated with $\boldsymbol{\omega}$, allowing $\boldsymbol{\omega}$ to be used to balance the relative magnitudes of the eigenvalues of the NTK across different tasks, thereby balancing the convergence speeds of different tasks.

Motivated by this, our method is designed based on the following strategy: In each training iteration, we compute the maximum eigenvalue $\lambda_i$ of the NTK matrix $\mathcal{K}_{ii}$ for task $i$, which serves as a good indicator of its current convergence speed. We then derive the task weights $\{\omega_i\}_{i=1}^{k}$ by normalizing these eigenvalues $\{\lambda_i\}_{i=1}^{k}$. Proposition 3.3 provides the most intuitive physical interpretation: With the introduction of $\omega_i$, the new NTK matrix for each task becomes $\omega_i^2 K_{ii}$, and its maximum eigenvalue is accordingly scaled to $\omega_i^2 \lambda_i$. Therefore, to achieve eigenvalue normalization across tasks, we enforce the condition that:

$$\omega_i \propto \frac{1}{\sqrt{\lambda_i}}, i = 1, \ldots, k. \tag{14}$$

Directly employing $\omega_i = \frac{1}{\sqrt{\lambda_i}}$ disregards the original scaling inherent in the task eigenvalues. Consequently, to retain the scale information reflecting the tasks' current convergence speeds, we utilize the average maximum eigenvalue, $\tilde{\lambda} = \frac{1}{k} \sum_{j=1}^{k} \lambda_j$, to scale $\{w_i\}_{i=1}^{k}$. This motivates the following definition for $\omega_i$:

$$\omega_i = \sqrt{\frac{\tilde{\lambda}}{\lambda_i}}, \ i = 1, \ldots, k. \tag{15}$$

The overall algorithm flow is shown in Algorithm 1. Our design of $\boldsymbol{\omega}$ enables the effective balancing of convergence speeds across different tasks by balancing the maximum eigenvalues of their NTK matrices, while preserving the scale information of the original eigenvalues. Nevertheless, computing the NTK matrix can be time-consuming since it requires dividing a batch into $n$ mini-batches and computing their gradients. This limitation has prompted us to find a way to compute the NTK with minimal cost, thereby extending the applicability of our method to more scenarios. The following section provides the solution.

| **Algorithm 1** NTKMTL | **Algorithm 2** NTKMTL-SR |
|---|---|
| 1: **Input:** Initial model parameters $\theta_0$; Learning rate $\{\eta_t\}$; number $n$ of mini batches. | 1: **Input:** Initial model parameters $\theta_0$; Learning rate $\{\eta_t\}$; number $n$ of mini batches. |
| 2: **for** $t = 0$ **to** $T - 1$ **do** | 2: **for** $t = 0$ **to** $T - 1$ **do** |
| 3: Compute $[J_1^t, \cdots, J_k^t]$ for $n$ mini-batches, and obtain the gradients $[g_1^t, \cdots, g_k^t]$. | 3: Compute $[J_1^t(z), \cdots, J_k^t(z)]$ with respect to $z$ for $n$ mini batches. |
| 4: Obtain the NTK $\{\mathcal{K}_{ii}^t\}_{i=1}^k$ and $\widetilde{\mathcal{K}^t}$ through Eq. 8 and 9. | 4: Obtain $\{\mathcal{K}_{ii}^t(z)\}_{i=1}^k$ and $\widetilde{\mathcal{K}^t}(z)$ through Eq.9 and 17. |
| 5: Compute $\{\omega_i\}_{i=1}^k$ through Eq. 15. | 5: Compute $\{\omega_i\}_{i=1}^k$ through Eq. 18. |
| 6: Obtain the update vector $g_s^t = \sum_{i=1}^k \omega_i g_i^t$. | 6: Obtain aggregated loss $\mathcal{L} = \sum_{i=1}^k \omega_i \ell_i$. |
| 7: Update $\theta_{t+1} = \theta_t - \eta_t g_s^t$. | 7: Back propagate and update parameters $\theta_{t+1}$. |
| 8: **end for** | 8: **end for** |

## 3.4  Approximation via Shared Representation

In MTL, the model typically consists of shared parameters $\theta$ and task-specific parameters for $k$ tasks, where the number of parameters in the task-specific components (typically 1–2 layers of linear or convolutional layers) is much smaller than the number of shared parameters. We define the shared representation $z$ as the output of the input $\mathbf{x}$ after passing through the shared parameters $\theta$. Then, by applying the chain rule, we can derive the following:

$$\frac{\partial f_i(\theta, \mathbf{x})}{\partial \theta} = \frac{\partial f_i(\theta, \mathbf{x})}{\partial z} \cdot \frac{\partial z}{\partial \theta}. \tag{16}$$

Note that $\frac{\partial z}{\partial \theta}$ is the same for all tasks and acts on all $\{f_i\}_{i=1}^k$. As a result, it further impacts the overall NTK $\widetilde{\mathcal{K}}$. This implies that $\widetilde{\mathcal{K}}$ can be viewed as a projection of the NTK matrix $\widetilde{\mathcal{K}}(z)$ of $z$ onto the feature space through $\frac{\partial z}{\partial \theta}$. These analyses indicate that, similar to some previous works [23, 47], NTKMTL has the ability to accelerate computation using shared representation, and we name this approximation algorithm NTKMTL-SR. Specifically, we consider approximating the original method using the NTK analysis of $z$, i,e. by replacing Eq. 8 with the following expression:

$$(\mathcal{K}_{ij}(z))_{uv} = \left\langle \frac{\partial f_i(\theta, x_u)}{\partial z}, \frac{\partial f_j(\theta, x_v)}{\partial z} \right\rangle, \tag{17}$$

which further leads to the subsequent formulation for $\boldsymbol{\omega}$:

$$\omega_i = \sqrt{\frac{\tilde{\lambda}(z)}{\lambda_i(z)}}, \; i = 1, \ldots, k, \tag{18}$$

where $\lambda_i(z)$ represents the maximum eigenvalue of the NTK matrix $K_{ii}(z)$, and $\tilde{\lambda}(z)$ is the average of these eigenvalues across all $k$ tasks, defined as $\tilde{\lambda}(z) = \frac{1}{k} \sum_{j=1}^k \lambda_j(z)$. Since computing the gradient of $f_i(\theta, \mathbf{x})$ with respect to $z$ only requires backpropagation through the task-specific parameters, it incurs little additional time and memory cost. After obtaining $\boldsymbol{\omega}$, instead of computing the gradients separately for each of the $k$ tasks and then weighting them, we can directly compute the aggregated loss using Eq. 12 and perform one backpropagation, meaning that we only need to compute the gradient for shared parameters $\theta$ once per iteration.

The overall algorithm is outlined in Algorithm 2. Fig. 1 illustrates the training speed on the CelebA benchmark with up to 40 tasks, showing that NTKMTL-SR exhibits nearly the same training speed as traditional loss-oriented Linear Scalarization. This further enhances the generalizability of our method under various constraints.

## 4  Experiments

### 4.1  Protocols

We evaluate the performance of our proposed NTKMTL and NTKMTL-SR across a wide range of MTL scenarios, including multi-task supervised learning and multi-task reinforcement learning. For

Table 1: Results on NYU-v2 (3-task) dataset. Each experiment is repeated 3 times with different random seeds and the average is reported. The detailed standard error is reported in the Appendix. The best scores are reported in gray.

| METHOD | SEGMENTATION | | DEPTH | | SURFACE NORMAL | | | | | MR ↓ | Δm% ↓ |
|---|---|---|---|---|---|---|---|---|---|---|---|
| | | | | | ANGLE DISTANCE ↓ | | WITHIN $t°$ ↑ | | | | |
| | MIOU ↑ | PIX ACC ↑ | ABS ERR ↓ | REL ERR ↓ | MEAN | MEDIAN | 11.25 | 22.5 | 30 | | |
| STL | 38.30 | 63.76 | 0.6754 | 0.2780 | 25.01 | 19.21 | 30.14 | 57.20 | 69.15 | | |
| LS | 39.29 | 65.33 | 0.5493 | 0.2263 | 28.15 | 23.96 | 22.09 | 47.50 | 61.08 | 17.44 | 5.59 |
| SI | 38.45 | 64.27 | 0.5354 | 0.2201 | 27.60 | 23.37 | 22.53 | 48.57 | 62.32 | 16.11 | 4.39 |
| RLW [28] | 37.17 | 63.77 | 0.5759 | 0.2410 | 28.27 | 24.18 | 22.26 | 47.05 | 60.62 | 20.22 | 7.78 |
| DWA [35] | 39.11 | 65.31 | 0.5510 | 0.2285 | 27.61 | 23.18 | 24.17 | 50.18 | 62.39 | 16.33 | 3.57 |
| UW [25] | 36.87 | 63.17 | 0.5446 | 0.2260 | 27.04 | 22.61 | 23.54 | 49.05 | 63.65 | 16.00 | 4.05 |
| MGDA [46] | 30.47 | 59.90 | 0.6070 | 0.2555 | 24.88 | 19.45 | 29.18 | 56.88 | 69.36 | 11.44 | 1.38 |
| PCGRAD [58] | 38.06 | 64.64 | 0.5550 | 0.2325 | 27.41 | 22.80 | 23.86 | 49.83 | 63.14 | 16.89 | 3.97 |
| GRADDROP [13] | 39.39 | 65.12 | 0.5455 | 0.2279 | 27.48 | 22.96 | 23.38 | 49.44 | 62.87 | 15.56 | 3.58 |
| CAGRAD [32] | 39.79 | 65.49 | 0.5486 | 0.2250 | 26.31 | 21.58 | 25.61 | 52.36 | 65.58 | 11.56 | 0.20 |
| IMTL-G [33] | 39.35 | 65.60 | 0.5426 | 0.2256 | 26.02 | 21.19 | 26.20 | 53.13 | 66.24 | 10.89 | -0.76 |
| MoCo [18] | 40.30 | 66.07 | 0.5575 | 0.2135 | 26.67 | 21.83 | 25.61 | 51.78 | 64.85 | 10.89 | 0.16 |
| NASH-MTL [39] | 40.13 | 65.93 | 0.5261 | 0.2171 | 25.26 | 20.08 | 28.40 | 55.47 | 68.15 | 8.00 | -4.04 |
| ALIGNED-MTL [47] | 40.15 | 66.05 | 0.5520 | 0.2291 | 25.37 | 19.89 | 28.30 | 55.29 | 67.95 | 10.44 | -3.12 |
| FAMO [31] | 38.88 | 64.90 | 0.5474 | 0.2194 | 25.06 | 19.57 | 29.21 | 56.61 | 68.98 | 9.00 | -4.10 |
| SDMGRAD [54] | 40.47 | 65.90 | 0.5225 | 0.2084 | 25.07 | 19.99 | 28.54 | 55.74 | 68.53 | 6.11 | -4.84 |
| DB-MTL [27] | 41.42 | 66.45 | 0.5251 | 0.2160 | 25.03 | 19.50 | 28.72 | 56.17 | 68.73 | 4.56 | -5.36 |
| STCH [30] | 41.35 | 66.07 | 0.4965 | 0.2010 | 26.55 | 21.81 | 24.84 | 51.39 | 64.86 | 8.22 | -1.35 |
| FAIRGRAD [4] | 39.74 | 66.01 | 0.5377 | 0.2236 | 24.84 | 19.60 | 29.26 | 56.58 | 69.16 | 6.44 | -4.66 |
| GO4ALIGN [48] | 40.42 | 65.37 | 0.5492 | 0.2167 | 24.76 | 18.94 | 30.54 | 57.87 | 69.84 | 5.00 | -6.08 |
| NTKMTL | 39.68 | 65.43 | 0.5296 | 0.2168 | 24.24 | 18.63 | 30.74 | 58.72 | 70.78 | 4.33 | -6.99 |
| NTKMTL-SR | 40.23 | 65.28 | 0.5261 | 0.2136 | 24.88 | 19.58 | 29.53 | 56.67 | 69.08 | 5.56 | -5.35 |

multi-task supervised learning, experiments are conducted on several benchmarks, including dense prediction tasks on the NYUv2 [49] and CityScapes [15] datasets, regression tasks on the QM9 [8] dataset, and image-level classification on the CelebA [36] dataset. For multi-task reinforcement learning, experiments are performed in the MT10 environment from the Meta-World benchmark [59]. Due to the memory and time cost associated with fully computing the gradients of $n$ mini-batches for shared parameters to construct the NTK matrix, we set $n = 1$ for NTKMTL to ensure fairness when comparing with other methods. In this case, it is consistent with other gradient-oriented methods, requiring $k$ gradient backpropagations per iteration. As for NTKMTL-SR, since its cost of constructing the NTK matrix is minimal, we set $n = 4$ by default based on experimental validation.

**Baselines.** We comprehensively compare the proposed NTKMTL and NTKMTL-SR with the following methods: Single-task learning (STL), Linear Scalarization (LS), Scale-Invariant (SI), Dynamic Weight Average (DWA) [35], Uncertainty Weighting (UW) [25], Multi-Gradient Descent Algorithm (MGDA) [46], Random Loss Weighting (RLW) [28], PCGrad [58], GradDrop [13], CAGrad [32], IMTL-G [33], Nash-MTL [39], Moco [18], Aligned-MTL [47], SDMGrad [54], DB-MTL [27], STCH [30], FAMO [31], FairGrad [4] and GO4Align [48].

**Metrics.** We follow previous works [39, 4] and use two overall performance metrics for MTL: (1) $\Delta m\%$, the **average performance drop** relative to the STL baseline: $\Delta m\% = \frac{1}{\mathcal{S}} \sum_{i=1}^{\mathcal{S}} (-1)^{\delta_i} \frac{(M_{m,i} - M_{b,i})}{M_{b,i}} \times 100\%$, where $\mathcal{S}$ is the number of metrics. $M_{b,i}$ is the baseline STL metric and $M_{m,i}$ is the metric from the MTL method. $\delta_i = 1$ if a higher value is better for $M_i$, and 0 otherwise. (2) **Mean Rank (MR)**: MR reports the average rank of a method across all tasks, where a lower value indicates better performance. A method with the top rank in all tasks has an MR of 1.

## 4.2 Multi-Task Supervised Learning

**Dense Prediction.** Widely used benchmarks in this domain include NYUv2 [49] and CityScapes [15]. The NYUv2 dataset includes three tasks: semantic segmentation, depth estimation, and surface normal prediction, while CityScapes includes semantic segmentation and depth estimation tasks. The experimental results are presented in Table 1 and 2 in the main text, and Table 7 in the Appendix.

On the NYUv2 dataset, the difficulty levels of the three tasks show significant variation. Previous methods generally outperform the Single Task Learning (STL) baseline in the tasks of semantic segmentation and depth estimation, but almost all of them consistently underperform STL on the surface normal prediction task, leading to a significant task imbalance in the overall results. In contrast, by leveraging NTK theory to balance convergence speed of each task during training, both NTKMTL and NTKMTL-SR show strong performance in the surface normal prediction task. Notably,

among all existing methods, only NTKMTL and GO4Align consistently outperform the STL baseline across all three tasks, achieving a more balanced optimization. Moreover, NTKMTL achieves SOTA on this benchmark with an impressive mean rank of **4.33** and the best performance drop of **-6.99%**.

On the CityScapes dataset, NTKMTL and NTKMTL-SR also perform exceptionally well, as shown in Table 2. More detailed results can be found in Table 7 in the Appendix. Compared to existing methods that tend to prioritize the optimization of semantic segmentation, NTKMTL and NTKMTL-SR achieve more balanced results across both segmentation and depth estimation tasks, with NTKMTL also obtaining the best performance drop.

**Image-Level Classification.** We also evaluated the performance of our proposed NTKMTL and NTKMTL-SR on the CelebA dataset. CelebA [36] is a large-scale facial attribute dataset consisting of over 200K images, each labeled with 40 attributes, such as smiling, wavy hair, and mustache. This task represents a 40-task MTL classification problem, where each task is designed to predict one binary attribute. This benchmark tests both the collaborative optimization capability and efficiency of MTL methods when dealing with a large number of tasks. The results are shown in Table 2. In this challenging setting, NTKMTL achieves better average performance than the STL baseline, as indicated by the negative $\Delta m\%$, which was not achievable by previous methods. NTKMTL also achieved state-of-the-art performance in both MR and $\Delta m\%$.

Table 2: Results on CityScapes (2-task) and CelebA (40-task). Each experiment is repeated 3 times with different random seeds and the average is reported. Detailed standard error is reported in the Appendix. Best scores are reported in gray.

| METHOD | CITYSCAPES | | CELEBA | |
|---|---|---|---|---|
| | MR ↓ | $\Delta m\% ↓$ | MR ↓ | $\Delta m\% ↓$ |
| LS | 8.25 | 22.60 | 7.85 | 4.15 |
| SI | 11.50 | 14.11 | 9.75 | 7.20 |
| RLW [28] | 10.25 | 24.38 | 6.90 | 1.46 |
| DWA [35] | 7.75 | 21.45 | 8.72 | 3.20 |
| UW [25] | 7.75 | 5.89 | 7.38 | 3.23 |
| MGDA [46] | 12.00 | 44.14 | 12.97 | 14.85 |
| PCGRAD [58] | 8.50 | 18.29 | 8.53 | 3.17 |
| CAGRAD [32] | 7.25 | 11.64 | 8.10 | 2.48 |
| IMTL-G [33] | 5.50 | 11.10 | 6.25 | 0.84 |
| NASH-MTL [39] | 3.50 | 6.82 | 6.50 | 2.84 |
| FAMO [31] | 7.75 | 8.13 | 6.45 | 1.21 |
| FAIRGRAD [4] | 2.25 | 5.18 | 6.92 | 0.37 |
| NTKMTL | 7.00 | **1.92** | 4.35 | **-0.77** |
| NTKMTL-SR | 6.25 | 3.84 | **4.33** | 0.23 |

Fig. 1 visualizes the training time per epoch for various loss-oriented and gradient-oriented methods. On the CelebA dataset with up to 40 tasks, NTKMTL-SR maintains a speed comparable to loss-oriented methods.

**Multi-Task Regression.** QM9 [8] is a widely used benchmark for multi-task regression, containing over 130K organic molecules represented as graphs. It includes 11 tasks, each requiring the prediction of a molecular property. Due to the large number of tasks, as well as the

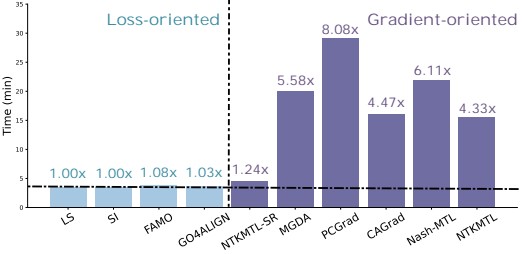

Figure 1: Training time per epoch for various methods on CelebA (40-task) dataset.

significant differences in task difficulty and convergence speeds, existing MTL methods exhibit a substantial performance drop compared to the STL baseline.

On this benchmark, previous methods were implemented using a shared MPNN codebase [39, 31, 19]. However, we find that the hyperparameter settings in this codebase are suboptimal (specifically, the improper learning rate scheduler makes the learning rate decay too quickly, resulting in incomplete convergence), potentially leading to unfair comparisons. Therefore, we adjusted specific hyperparameters and reproduced all baselines for a fair comparison. To ensure accurate evaluation, we also reproduced the 11-task STL baselines under the same settings.

Experimental results are presented in Table 3. More detailed explanations and settings are provided in Appendix C. As both STL baselines and MTL methods show significant performance improvements under the new settings, we can observe different behaviors among previous methods. The final $\Delta m\%$ of some methods (e.g., LS, RLW, PCGrad, CAGrad) were largely consistent with their originally reported values. For other methods (e.g., SI, UW, FAMO, GO4Align), the $\Delta m\%$ significantly improved compared to the reported results. This indicates that the previous hyperparameter settings fail to fully exhibit the capabilities of these methods. Under the new settings that better ensure convergence, their performance shows further improvement. On this benchmark, NTKMTL shows competitive results. NTKMTL-SR surpasses NTKMTL in performance and achieves state-of-the-art

Table 3: Results on QM9 (11-task) dataset. All baselines are reproduced under optimized hyperparameter settings. The best scores are reported in  gray . More details are reported in the Appendix.

| METHOD | $\mu$ | $\alpha$ | $\epsilon_{HOMO}$ | $\epsilon_{LUMO}$ | $\langle R^2 \rangle$ | ZPVE | $U_0$ | $U$ | $H$ | $G$ | $c_v$ | MR↓ | $\Delta m\%$↓ |
|---|---|---|---|---|---|---|---|---|---|---|---|---|---|
| | | | | | | MAE↓ | | | | | | | |
| STL | 0.060 | 0.156 | 60.54 | 51.22 | 0.419 | 3.08 | 39.3 | 42.9 | 41.7 | 43.1 | 0.061 | | |
| LS | 0.077 | 0.253 | 55.95 | 68.59 | 4.163 | 11.08 | 109.2 | 109.8 | 110.1 | 106.7 | 0.099 | 9.82 | 179.8 |
| SI | 0.159 | 0.242 | 109.6 | 96.80 | 0.732 | 3.308 | 34.35 | 34.37 | 34.33 | 35.16 | 0.081 | 5.27 | 39.7 |
| RLW [28] | 0.090 | 0.277 | 62.49 | 76.39 | 4.948 | 12.54 | 124.8 | 124.9 | 125.0 | 122.1 | 0.115 | 12.09 | 222.6 |
| DWA [35] | 0.078 | 0.239 | 55.17 | 67.40 | 3.992 | 10.92 | 107.4 | 108.1 | 108.3 | 105.4 | 0.098 | 8.73 | 173.0 |
| UW [25] | 0.194 | 0.274 | 120.6 | 102.8 | 0.763 | 3.698 | 41.11 | 41.13 | 41.16 | 41.75 | 0.089 | 8.27 | 58.4 |
| MGDA [46] | 0.154 | 0.266 | 95.20 | 67.51 | 3.088 | 4.468 | 49.38 | 49.21 | 49.62 | 49.69 | 0.087 | 8.36 | 101.4 |
| PCGRAD [58] | 0.078 | 0.221 | 59.14 | 67.82 | 2.937 | 6.691 | 88.24 | 88.65 | 88.85 | 87.36 | 0.084 | 7.91 | 118.6 |
| CAGRAD [32] | 0.083 | 0.234 | 57.80 | 70.98 | 2.718 | 5.352 | 76.47 | 76.93 | 77.05 | 76.32 | 0.089 | 8.27 | 102.4 |
| NASH-MTL [39] | 0.086 | 0.218 | 69.78 | 66.18 | 2.153 | 4.679 | 59.63 | 59.94 | 59.98 | 59.97 | 0.082 | 6.91 | 72.9 |
| FAMO [31] | 0.128 | 0.230 | 98.09 | 84.42 | 0.859 | 3.541 | 40.24 | 40.57 | 40.62 | 40.21 | 0.081 | 5.91 | 38.9 |
| FAIRGRAD [4] | 0.109 | 0.208 | 81.74 | 72.82 | 1.669 | 3.418 | 51.31 | 51.67 | 51.72 | 51.97 | 0.079 | 6.64 | 57.0 |
| GO4ALIGN [48] | 0.113 | 0.314 | 74.46 | 91.04 | 0.912 | 3.632 | 36.06 | 36.38 | 36.41 | 36.58 | 0.104 | 6.64 | 40.5 |
| NTKMTL | 0.091 | 0.212 | 70.97 | 70.81 | 2.113 | 3.835 | 44.18 | 44.56 | 44.53 | 44.38 | 0.077 | 5.91 | 56.7 |
| NTKMTL-SR | 0.081 | 0.207 | 75.95 | 69.10 | 1.176 | 3.689 | 40.14 | 40.46 | 40.48 | 40.49 | 0.074 | 4.00 | 30.7 |

results, which we attribute to the natural alignment of the L2 loss used in regression tasks as analyzed in Appendix B.1. As a result, larger $n$ produces a more accurate estimate of convergence speed.

## 4.3 Multi-Task Reinforcement Learning

We further assess our method using the MT10 benchmark, which consists of 10 robotic manipulation tasks from the MetaWorld environment [59]. The goal in this setting is to train a single policy that can generalize across a variety of tasks, including pick-and-place and door-opening. Specifically, we follow the setup outlined in previous works [39, 31] and use Soft Actor-Critic (SAC) [21] as the core algorithm. Our implementation builds upon the MTRL codebase from [39, 4], training the model for 2 million steps with a batch size of 1280.

In contrast to the multi-task supervised learning networks where shared and task-specific parameters can be easily distinguished, the MTRL problems commonly involve learning a single policy. Consequently, it is difficult to partition the parameters into shared and task-specific components in the same manner. Therefore, while applying NTKMTL-SR to this scenario is challenging, we primarily validate the performance of NTKMTL. We compare NTKMTL with several state-of-the-art methods, including Multi-task SAC (MTL SAC) [59], Multi-task SAC with Task Encoder (MTL SAC + TE) [59], Multi-headed SAC (MH SAC) [59], PCGrad [58], CAGrad [32], MoCo [18], Nash-MTL [39], FAMO [31], FairGrad [4] and Aligned-MTL [47]. The experimental results are reported in Table 4. In the MTRL scenario, NTKMTL continues to demonstrate strong performance with a competitive success rate.

Table 4: Results on MT10 benchmark across 10 random seeds.

| METHOD | SUCCESS RATE (MEAN ± STDERR) |
|---|---|
| STL | $0.90 \pm 0.03$ |
| MTL SAC [59] | $0.49 \pm 0.07$ |
| MTL SAC + TE [59] | $0.54 \pm 0.05$ |
| MH SAC [59] | $0.61 \pm 0.04$ |
| PCGRAD [58] | $0.72 \pm 0.02$ |
| CAGRAD [32] | $0.83 \pm 0.05$ |
| MoCo [18] | $0.75 \pm 0.05$ |
| NASH-MTL [39] | $0.91 \pm 0.03$ |
| FAMO [31] | $0.83 \pm 0.05$ |
| FAIRGRAD [4] | $0.84 \pm 0.07$ |
| ALIGNED-MTL [47] | $0.97 \pm 0.05$ |
| NTKMTL | $0.96 \pm 0.03$ |

## 5 Conclusion, Limitations and Future Work

In this paper, we introduce a new perspective on understanding task imbalance in MTL by leveraging NTK theory for analysis, and propose a new MTL method, NTKMTL. Specifically, we conduct spectral analysis of the NTK matrix during training, adjust the maximum eigenvalues of the task-specific NTK matrices to balance their convergence speeds, thereby mitigating task imbalance. Furthermore, we present NTKMTL-SR, an efficient variant based on approximation via shared representation, which achieves competitive performance with improved training efficiency. Extensive experiments have shown the strong performance of both NTKMTL and NTKMTL-SR, further demonstrating the applicability and generalization of our method across a wide range of scenarios.

**Limitations and Future Work.** In this work, we proposed the extended NTK $\widetilde{\mathcal{K}}$ for MTL as a tool for more comprehensive analysis of training dynamics across tasks. The weight design of our current method mainly focuses on the analysis of the task-specific NTK matrices $\mathcal{K}_{ii}$. The full structure

of the extended NTK matrix $\widetilde{\mathcal{K}}$ offers further analytical opportunities. For example, analyzing the off-diagonal $\mathcal{K}_{ij}$ blocks could reveal novel insights into task interactions or guide the development of task grouping strategies. These potential avenues are left for future investigation.

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

# A  Definitions of Notations

Due to the numerous concepts and theoretical derivations presented in the paper, we provide detailed definitions of the notations in Table 5 to assist readers in better understanding the content.

Table 5: Definitions of notations in this paper.

| Variable | Definition |
|---|---|
| $k$ | the number of tasks |
| $f$ | the mapping function of a deep neural network |
| $f_i$ | the mapping function of $i$-th task in MTL scenario |
| $n$ | the number of data points in the dataset, or the number of mini-batches in a batch |
| $(\mathbf{x}, \mathbf{y})$ | training dataset $\{(x_i, y_i)\}_{i=1}^n$ in Single-Task Learning (STL) setting |
| $\{\mathbf{y}_i\}_{i=1}^k$ | multiple labels for the $k$ tasks in MTL setting |
| $\theta$ | the network parameters, specifically referring to the shared parameters in MTL |
| $t$ | the current time step |
| $\mathcal{K}$ | the Neural Tangent Kernel matrix defined in Eq. 1 |
| $\lambda$ | the eigenvalues of the NTK |
| $\mathcal{L}$ | the overall loss function, with different definitions in different scenarios |
| $\{\ell_i\}_{i=1}^k$ | the losses with respect to $k$ tasks |
| $\mathcal{O}(t)$ | the output $\{f(x_i, \theta(t))\}_{i=1}^n$ of the network in STL setting and time $t$ |
| $\{\mathcal{O}_i(t)\}_{i=1}^k$ | the output of the network for $k$ tasks in MTL setting and time $t$ |
| $\mathbf{I}$ | the Identity matrix |
| $\eta_t$ | the learning rate at time $t$ |
| $\widetilde{\mathcal{K}}$ | the extended NTK matrix defined in Eq. 9 |
| $J_i$ | the Jacobian matrix of $f_i$ with respect to $\theta$ |
| $\boldsymbol{\omega}$ | $\{\omega_i\}_{i=1}^k$, representing the weights of the different tasks |
| $z$ | the shared representation of the input data in MTL setting |

# B  Theoretical Analysis

## B.1  Proof of Theorem 3.1

**Theorem 3.1.** *Let $\mathcal{O}(t) = \{f(\theta, x_i)\}_{i=1}^n$ be the outputs of the neural network at time $t$. $\mathbf{x} = \{x_i\}_{i=1}^n$ is the input data, and $\mathbf{y} = \{y_i\}_{i=1}^n$ is the corresponding label, Then $\mathcal{O}(t)$ follows this evolution:*

$$\frac{d\mathcal{O}(t)}{dt} = -\mathcal{K} \cdot (\mathcal{O}(t) - \mathbf{y}).$$

*Proof.* Following previous works [44, 52, 62], we build up the analysis framework for NTK in a supervised regression setting. The overall loss $\mathcal{L}(\theta)$ is defined as:

$$\mathcal{L}(\theta) = \sum_{i=1}^n \frac{1}{2}(f(\theta, x_i) - y_i)^2. \tag{19}$$

Through the gradient flow in Eq. 2, we can obtain:

$$
\begin{aligned}
\frac{d\theta}{dt} &= -\nabla_\theta \mathcal{L}(\theta) \\
&= -\sum_{i=1}^{n} \frac{\partial \mathcal{L}}{\partial f(\theta, x_i)} \cdot \frac{\partial f(\theta, x_i)}{\partial \theta} \\
&= -\sum_{i=1}^{n} \frac{\partial f(\theta, x_i)}{\partial \theta} (f(\theta, x_i) - y_i).
\end{aligned}
\tag{20}
$$

In fact, for the commonly used cross-entropy loss, the above equation also holds. In this case, considering a data point $x_j$, we have

$$
\begin{aligned}
\frac{df(\theta, x_j)}{dt} &= \frac{df(\theta, x_j)}{d\theta} \cdot \frac{d\theta}{dt} \\
&= \frac{df(\theta, x_j)}{d\theta} \left[ -\sum_{i=1}^{n} \frac{\partial f(\theta, x_i)}{\partial \theta} (f(\theta, x_i) - y_i) \right] \\
&= -\sum_{i=1}^{n} \left\langle \frac{df(\theta, x_j)}{d\theta}, \frac{\partial f(\theta, x_i)}{\partial \theta} \right\rangle (f(\theta, x_i) - y_i).
\end{aligned}
\tag{21}
$$

Given that $\mathcal{O}(t) = \{f(x_i, \theta(t))\}_{i=1}^{n}$ and $\mathbf{y} = \{y_i\}_{i=1}^{n}$, we can express Eq. 21 in vector form:

$$
\frac{d\mathcal{O}(t)}{dt} = -\mathcal{K} \cdot (\mathcal{O}(t) - \mathbf{y}),
$$

where $\mathcal{K}$ is defined as

$$
\mathcal{K}_{uv} = \left\langle \frac{\partial f(\theta, x_u)}{\partial \theta}, \frac{\partial f(\theta, x_v)}{\partial \theta} \right\rangle.
$$

Q.E.D.

**Discussions.** Another equivalent definition of NTK given the Jacobian matrix $J$ is:

$$
\mathcal{K} = JJ^\top.
\tag{22}
$$

This theorem shows that NTK connects the error term $\mathcal{O}(t) - \mathbf{y}$ to the changing rate of the output. Therefore, the theory can be used to analyze the training behaviors of neural networks. $\qquad\square$

### B.2 Proof of Theorem 3.2

**Theorem 3.2.** *Let $\{\mathcal{O}_1(t), \mathcal{O}_2(t), \ldots, \mathcal{O}_k(t)\}$ denote the outputs of the neural network function $\{f_1, f_2, \ldots, f_k\}$ for the $k$ task at time $t$, and let $\{\mathbf{y}_1, \mathbf{y}_2, \ldots, \mathbf{y}_k\}$ represent the corresponding labels. Then, the ordinary differential equation in Eq. 2 gives the following evolution:*

$$
\begin{bmatrix} \frac{d\mathcal{O}_1(t)}{dt} \\ \vdots \\ \frac{d\mathcal{O}_k(t)}{dt} \end{bmatrix} = - \underbrace{\begin{bmatrix} \mathcal{K}_{11} & \cdots & \mathcal{K}_{1k} \\ \vdots & \ddots & \vdots \\ \mathcal{K}_{k1} & \cdots & \mathcal{K}_{kk} \end{bmatrix}}_{\widetilde{\mathcal{K}}} \begin{bmatrix} \mathcal{O}_1(t) - \mathbf{y}_1 \\ \vdots \\ \mathcal{O}_k(t) - \mathbf{y}_k \end{bmatrix},
$$

*where $\mathcal{K}_{ij} \in \mathbb{R}^{n \times n}$ and $\mathcal{K}_{ij} = \mathcal{K}_{ji}^\top$ for $1 \le i, j \le k$. The $(u, v)$-th entry of $\mathcal{K}_{ij}$ is defined as*

$$
(\mathcal{K}_{ij})_{uv} = \left\langle \frac{\partial f_i(\theta, x_u)}{\partial \theta}, \frac{\partial f_j(\theta, x_v)}{\partial \theta} \right\rangle.
$$

*Proof.* In multi-task learning, a neural network with shared parameters $\theta$ is trained to simultaneously learn $k$ distinct tasks. In the general case, the overall loss function is defined as

$$
\mathcal{L}(\theta) = \sum_{i=1}^{k} \ell_i(\theta).
\tag{23}
$$

Similar to the derivation in Theorem 3.1, by utilizing the gradient flow in Eq. 2, we can derive:

$$
\begin{aligned}
\frac{d\theta}{dt} &= -\nabla_\theta \mathcal{L}(\theta) \\
&= -\sum_{i=1}^{k}\sum_{u=1}^{n} \frac{\partial \ell_i}{\partial f_i(\theta, x_u)} \cdot \frac{\partial f_i(\theta, x_u)}{\partial \theta} \\
&= -\sum_{i=1}^{k}\sum_{u=1}^{n} \frac{\partial f_i(\theta, x_u)}{\partial \theta} \cdot (f_i(\theta, x_u) - y_{i,u}),
\end{aligned}
\tag{24}
$$

where $y_{i,u}$ represents the ground truth label of the $u$-th element in the label set $\mathbf{y}_i$. For a data point $x_v$ and the function $f_j$ of task $j$, we have:

$$
\begin{aligned}
\frac{df_j(\theta, x_v)}{dt} &= \frac{df_j(\theta, x_v)}{d\theta} \cdot \frac{d\theta}{dt} \\
&= \frac{df_j(\theta, x_v)}{d\theta} \left[ -\sum_{i=1}^{k}\sum_{u=1}^{n} \frac{\partial f_i(\theta, x_u)}{\partial \theta} \cdot (f_i(\theta, x_u) - y_{i,u}) \right] \\
&= -\sum_{i=1}^{k}\sum_{u=1}^{n} \left\langle \frac{df_j(\theta, x_v)}{d\theta}, \frac{\partial f_i(\theta, x_u)}{\partial \theta} \right\rangle (f_i(\theta, x_u) - y_{i,u}).
\end{aligned}
\tag{25}
$$

Rewriting Eq. 25 in the form of high-dimensional vectors gives:

$$
\begin{bmatrix} \frac{d\mathcal{O}_1(t)}{dt} \\ \vdots \\ \frac{d\mathcal{O}_k(t)}{dt} \end{bmatrix} = - \underbrace{\begin{bmatrix} \mathcal{K}_{11} & \cdots & \mathcal{K}_{1k} \\ \vdots & \ddots & \vdots \\ \mathcal{K}_{k1} & \cdots & \mathcal{K}_{kk} \end{bmatrix}}_{\widetilde{\mathcal{K}}} \begin{bmatrix} \mathcal{O}_1(t) - \mathbf{y}_1 \\ \vdots \\ \mathcal{O}_k(t) - \mathbf{y}_k \end{bmatrix},
$$

where $\mathcal{K}_{ij} \in \mathbb{R}^{n \times n}$ and $\mathcal{K}_{ij} = \mathcal{K}_{ji}^\top$ for $1 \leq i, j \leq k$. The $(u, v)$-th entry of $\mathcal{K}_{ij}$ is defined as

$$
(\mathcal{K}_{ij})_{uv} = \left\langle \frac{\partial f_i(\theta, x_u)}{\partial \theta}, \frac{\partial f_j(\theta, x_v)}{\partial \theta} \right\rangle.
$$

Q.E.D.

**Discussions.** It can be observed that Theorem 3.2 essentially extends the foundational NTK theory from Theorem 3.1 to the MTL scenario. In this case, the input, output, and NTK matrix each acquire an additional dimension, corresponding to the task-oriented dimension. $\qquad\square$

## B.3   Proof of Proposition 3.3

**Proposition 3.3.   (Extension of Theorem 3.2)** *Let $\{\mathcal{O}_i(t)\}_{i=1}^{k}$, $\{\mathbf{y}_i\}_{i=1}^{k}$, and $\{\mathcal{K}_{ij}\}_{1 \leq i,j \leq k}$ be defined as in Theorem 3.2. We now replace the MTL optimization objective with the weighted form as presented in Eq. 12. Consequently, the ordinary differential equation governing the MTL training dynamics in Eq. 7 becomes:*

$$
\begin{bmatrix} \frac{d\mathcal{O}_1(t)}{dt} \\ \vdots \\ \frac{d\mathcal{O}_k(t)}{dt} \end{bmatrix} = - \underbrace{\begin{bmatrix} \omega_1^2 \mathcal{K}_{11} & \cdots & \omega_1 \omega_k \mathcal{K}_{1k} \\ \vdots & \ddots & \vdots \\ \omega_k \omega_1 \mathcal{K}_{k1} & \cdots & \omega_k^2 \mathcal{K}_{kk} \end{bmatrix}}_{\boldsymbol{\omega}\boldsymbol{\omega}^\top \odot \widetilde{\mathcal{K}}} \begin{bmatrix} \mathcal{O}_1(t) - \mathbf{y}_1 \\ \vdots \\ \mathcal{O}_k(t) - \mathbf{y}_k \end{bmatrix}.
\tag{26}
$$

*Proof.* Consider the weighted loss

$$
\mathcal{L}(\theta) = \sum_{i=1}^{k} \omega_i \ell_i(\theta),
\tag{27}
$$

the analysis in Appendix B.2 becomes

$$\frac{d\theta}{dt} = -\nabla_\theta \mathcal{L}(\theta)$$

$$= -\sum_{i=1}^{k} \sum_{u=1}^{n} \frac{\partial \omega_i \ell_i}{\partial f_i(\theta, x_u)} \cdot \frac{\partial f_i(\theta, x_u)}{\partial \theta} \tag{28}$$

$$= -\sum_{i=1}^{k} \omega_i \sum_{u=1}^{n} \frac{\partial f_i(\theta, x_u)}{\partial \theta} \cdot (f_i(\theta, x_u) - y_{i,u}).$$

Under the action of $\boldsymbol{\omega}$, the network output $\{\mathcal{O}_1, \dots, \mathcal{O}_k\}$ is transformed to $\{\omega_1 f_1, \dots, \omega_k f_k\}$. Consider the output of the network for the $j$-th task at a data point $x_v$:

$$\frac{d\omega_j f_j(\theta, x_v)}{dt} = \omega_j \frac{df_j(\theta, x_v)}{d\theta} \cdot \frac{d\theta}{dt}$$

$$= \omega_j \frac{df_j(\theta, x_v)}{d\theta} \left[ -\sum_{i=1}^{k} \omega_i \sum_{u=1}^{n} \frac{\partial f_i(\theta, x_u)}{\partial \theta} \cdot (f_i(\theta, x_u) - y_{i,u}) \right] \tag{29}$$

$$= -\sum_{i=1}^{k} \sum_{u=1}^{n} \omega_j \omega_i \left\langle \frac{df_j(\theta, x_v)}{d\theta}, \frac{\partial f_i(\theta, x_u)}{\partial \theta} \right\rangle (f_i(\theta, x_u) - y_{i,u}).$$

Then we have

$$\begin{bmatrix} \frac{d\mathcal{O}_1(t)}{dt} \\ \vdots \\ \frac{d\mathcal{O}_k(t)}{dt} \end{bmatrix} = - \underbrace{\begin{bmatrix} \omega_1^2 \mathcal{K}_{11} & \cdots & \omega_1 \omega_k \mathcal{K}_{1k} \\ \vdots & \ddots & \vdots \\ \omega_k \omega_1 \mathcal{K}_{k1} & \cdots & \omega_k^2 \mathcal{K}_{kk} \end{bmatrix}}_{\boldsymbol{\omega}\boldsymbol{\omega}^\top \odot \widetilde{\mathcal{K}}} \begin{bmatrix} \mathcal{O}_1(t) - \mathbf{y}_1 \\ \vdots \\ \mathcal{O}_k(t) - \mathbf{y}_k \end{bmatrix}$$

Q.E.D.

**Discussions.** Proposition 3.3 actually provides an intuition: by adjusting the relative magnitudes of $\{\omega_i\}_{i=1}^{k}$, one can alter the eigenvalue distribution of the NTK matrix, thereby balancing convergence speeds. This serves as the foundation for the design of NTKMTL and NTKMTL-SR. □

## C   Detailed Experimental Results

### C.1   Detailed Results on QM9

For QM9 benchmark, previous methods were all implemented based on a shared codebase [39, 31, 4], utilizing the message-passing neural network (MPNN) architecture [19]. All the methods were trained for 300 epochs. However, we found that the hyperparameter settings in this codebase were suboptimal (specifically, the improper learning rate scheduler makes the learning rate decay too quickly). **Consequently, the reported results of most previous methods had not fully converged, potentially leading to unfair comparisons.** Therefore, we readjusted the hyperparameters: **we changed the batch size from 120 to 60 and the learning rate scheduler's patience from 5 to 10. The remaining hyperparameters were kept unchanged, consistent with the original codebase.** Subsequently, we reproduced all baselines that had reported results on QM9, and their performance on various tasks significantly improved under these revised settings. To ensure accurate evaluation, we also reproduced the 11-task STL baselines under the same settings.

Experimental results are presented in Table 6. For each baseline, the upper row shows results taken from its original paper, while the bottom row (highlighted in green) presents results reproduced by us under new hyperparameter settings. It can be seen that without changing the model architecture, solely by adjusting the learning rate schedule, the performance of all baselines on the 11 tasks has significantly improved, notably extending the Pareto front.

Under the new settings, with both STL baselines and MTL methods showing significant performance improvements, we observed different behaviors among previous methods. The final $\Delta m\%$ of some methods (e.g., LS, RLW, PCGrad, CAGrad) were largely consistent with their originally reported values. For other methods (e.g., SI, UW, FAMO, GO4Align), the $\Delta m\%$ significantly improved

Table 6: Results on QM9 (11-task) dataset. For each baseline, the upper row shows results taken from its original paper, while the bottom row (highlighted in **green**) presents results reproduced by us under new hyperparameter settings.

| METHOD | $\mu$ | $\alpha$ | $\epsilon_{HOMO}$ | $\epsilon_{LUMO}$ | $\langle R^2 \rangle$ | ZPVE | $U_0$ | $U$ | $H$ | $G$ | $c_v$ | MR↓ | $\Delta m\%$↓ |
|---|---|---|---|---|---|---|---|---|---|---|---|---|---|
| | | | | | | MAE↓ | | | | | | | |
| STL | 0.067 | 0.181 | 60.57 | 53.91 | 0.502 | 4.53 | 58.8 | 64.2 | 63.8 | 66.2 | 0.072 | | |
| STL | 0.060 | 0.156 | 60.54 | 51.22 | 0.419 | 3.08 | 39.3 | 42.9 | 41.7 | 43.1 | 0.061 | | |
| LS | 0.106 | 0.325 | 73.57 | 89.67 | 5.19 | 14.06 | 143.4 | 144.2 | 144.6 | 140.3 | 0.128 | | 177.6 |
| LS | 0.077 | 0.253 | 55.95 | 68.59 | 4.163 | 11.08 | 109.2 | 109.8 | 110.1 | 106.7 | 0.099 | 9.82 | 179.8 |
| SI | 0.309 | 0.345 | 149.8 | 135.7 | 1.00 | 4.50 | 55.3 | 55.75 | 55.82 | 55.27 | 0.112 | | 77.8 |
| SI | 0.159 | 0.242 | 109.6 | 96.80 | 0.732 | 3.308 | 34.35 | 34.37 | 34.33 | 35.16 | 0.081 | 5.27 | 39.7 |
| RLW [28] | 0.113 | 0.340 | 76.95 | 92.76 | 5.86 | 15.46 | 156.3 | 157.1 | 157.6 | 153.0 | 0.137 | | 203.8 |
| RLW [28] | 0.090 | 0.277 | 62.49 | 76.39 | 4.948 | 12.54 | 124.8 | 124.9 | 125.0 | 122.1 | 0.115 | 12.09 | 222.6 |
| DWA [35] | 0.107 | 0.325 | 74.06 | 90.61 | 5.09 | 13.99 | 142.3 | 143.0 | 143.4 | 139.3 | 0.125 | | 175.3 |
| DWA [35] | 0.078 | 0.239 | 55.17 | 67.40 | 3.992 | 10.92 | 107.4 | 108.1 | 108.3 | 105.4 | 0.098 | 8.73 | 173.0 |
| UW [25] | 0.386 | 0.425 | 166.2 | 155.8 | 1.06 | 4.99 | 66.4 | 66.78 | 66.80 | 66.24 | 0.122 | | 108.0 |
| UW [25] | 0.194 | 0.274 | 120.6 | 102.8 | 0.763 | 3.698 | 41.11 | 41.13 | 41.16 | 41.75 | 0.089 | 8.27 | 58.4 |
| MGDA [46] | 0.217 | 0.368 | 126.8 | 104.6 | 3.22 | 5.69 | 88.37 | 89.4 | 89.32 | 88.01 | 0.120 | | 120.5 |
| MGDA [46] | 0.154 | 0.266 | 95.20 | 67.51 | 3.088 | 4.468 | 49.38 | 49.21 | 49.62 | 49.69 | 0.087 | 8.36 | 101.4 |
| PCGRAD [58] | 0.106 | 0.293 | 75.85 | 88.33 | 3.94 | 9.15 | 116.36 | 116.8 | 117.2 | 114.5 | 0.110 | | 125.7 |
| PCGRAD [58] | 0.078 | 0.221 | 59.14 | 67.82 | 2.937 | 6.691 | 88.24 | 88.65 | 88.85 | 87.36 | 0.084 | 7.91 | 118.6 |
| CAGRAD [32] | 0.118 | 0.321 | 83.51 | 94.81 | 3.21 | 6.93 | 113.99 | 114.3 | 114.5 | 112.3 | 0.116 | | 112.8 |
| CAGRAD [32] | 0.083 | 0.234 | 57.80 | 70.98 | 2.718 | 5.352 | 76.47 | 76.93 | 77.05 | 76.32 | 0.089 | 8.27 | 102.4 |
| NASH-MTL [39] | 0.102 | 0.248 | 82.95 | 81.89 | 2.42 | 5.38 | 74.5 | 75.02 | 75.10 | 74.16 | 0.093 | | 62.0 |
| NASH-MTL [39] | 0.086 | 0.218 | 69.78 | 66.18 | 2.153 | 4.679 | 59.63 | 59.94 | 59.98 | 59.97 | 0.082 | 6.91 | 72.9 |
| FAMO [31] | 0.15 | 0.30 | 94.0 | 95.2 | 1.63 | 4.95 | 70.82 | 71.2 | 71.2 | 70.3 | 0.10 | | 58.5 |
| FAMO [31] | 0.128 | 0.230 | 98.09 | 84.42 | 0.859 | 3.541 | 40.24 | 40.57 | 40.62 | 40.21 | 0.081 | 5.91 | 38.9 |
| FAIRGRAD [4] | 0.117 | 0.253 | 87.57 | 84.00 | 2.15 | 5.07 | 70.89 | 71.17 | 71.21 | 70.88 | 0.095 | | 57.9 |
| FAIRGRAD [4] | 0.109 | 0.208 | 81.74 | 72.82 | 1.669 | 3.418 | 51.31 | 51.67 | 51.72 | 51.97 | 0.079 | 6.64 | 57.0 |
| GO4ALIGN [48] | 0.17 | 0.35 | 102.4 | 119.0 | 1.22 | 4.94 | 53.9 | 54.3 | 54.3 | 53.9 | 0.11 | | 52.7 |
| GO4ALIGN [48] | 0.113 | 0.314 | 74.46 | 91.04 | 0.912 | 3.632 | 36.06 | 36.38 | 36.41 | 36.58 | 0.104 | 6.64 | 40.5 |
| NTKMTL | 0.091 | 0.212 | 70.97 | 70.81 | 2.113 | 3.835 | 44.18 | 44.56 | 44.53 | 44.38 | 0.077 | 5.91 | 56.7 |
| NTKMTL-SR | 0.081 | 0.207 | 75.95 | 69.10 | 1.176 | 3.689 | 40.14 | 40.46 | 40.48 | 40.49 | 0.074 | 4.00 | 30.7 |

compared to the reported results. This indicates that the previous hyperparameter settings fail to fully exhibit the capabilities of these methods. Under the new settings that better ensure convergence, their performance shows further improvement. Notably, the traditional Scale-Invariant Linear Scalarization (SI) demonstrated extremely superior performance, surpassing the vast majority of recent baselines. This suggests that in cases where differences in loss scales among different tasks are too large, directly eliminating scale differences through logarithmic methods may be an effective solution.

In summary, we identified that the parameter settings in the previous code implementation hindered the smooth convergence of both STL and MTL methods, rendering comparisons made under these conditions unfair. Our experiments verified that this issue can be resolved by simply adjusting the batch size and learning rate scheduler parameters. Under the new settings that better ensure convergence, the performance of both STL methods and numerous MTL methods on the 11 tasks has significantly improved. **We believe that fair and transparent reproduction of all previous baseline methods under such parameter settings enables more reasonable comparisons on this benchmark, providing new results that are valuable to the MTL community.**

## C.2 Detailed Results with Standard Errors

In this section, We provide the detailed experimental results with standard errors for our method, and the results for the baseline methods are taken from their original papers. Since results for CelebA are not reported by some methods [47, 54, 48], these methods are excluded when presenting combined CityScapes and CelebA results in Table 2. Table 7 provides detailed results solely on the CityScapes dataset, including these methods. Consequently, the mean rank (MR) in Table 7 slightly differs from that in Table 2.

On NYUv2 and CityScapes, we follow the training settings of [39, 4], including data augmentation for all compared methods. Training runs for 200 epochs, with the learning rate initialized at $10^{-4}$ and reduced to $5 \times 10^{-5}$ after 100 epochs. The architecture is the SegNet-based [3] Multi-Task Attention Network (MTAN) [33]. Batch sizes are 2 (NYUv2) and 8 (CityScapes), and the hyparameter $n$

Table 7: Detailed results on CityScapes (2-task) dataset. Each experiment is repeated 3 times with different random seeds and the average is reported. The best scores are reported in gray.

| | CITYSCAPES | | | | | |
|---|---|---|---|---|---|---|
| METHOD | SEGMENTATION | | DEPTH | | MR ↓ | Δm% ↓ |
| | mIoU ↑ | PIX ACC ↑ | ABS ERR ↓ | REL ERR ↓ | | |
| STL | 74.01 | 93.16 | 0.0125 | 27.77 | | |
| LS | 75.18 | 93.49 | 0.0155 | 46.77 | 10.25 | 22.60 |
| SI | 70.95 | 91.73 | 0.0161 | 33.83 | 14.00 | 14.11 |
| RLW [28] | 74.57 | 93.41 | 0.0158 | 47.79 | 12.25 | 24.38 |
| DWA [35] | 75.24 | 93.52 | 0.0160 | 44.37 | 9.75 | 21.45 |
| UW [25] | 72.02 | 92.85 | 0.0140 | 30.13 | 10.00 | 5.89 |
| MGDA [46] | 68.84 | 91.54 | 0.0309 | 33.50 | 14.75 | 44.14 |
| PCGRAD [58] | 75.13 | 93.48 | 0.0154 | 42.07 | 10.50 | 18.29 |
| CAGRAD [32] | 75.16 | 93.48 | 0.0141 | 37.60 | 9.25 | 11.64 |
| IMTL-G [33] | 75.33 | 93.49 | 0.0135 | 38.41 | 7.25 | 11.10 |
| NASH-MTL [39] | 75.41 | 93.66 | 0.0129 | 35.02 | 4.75 | 6.82 |
| FAMO [31] | 74.54 | 93.29 | 0.0145 | 32.59 | 9.25 | 8.13 |
| ALIGNED-MTL [47] | 75.77 | 93.69 | 0.0133 | 32.66 | 3.00 | 5.27 |
| SDMGRAD [54] | 74.53 | 93.52 | 0.0137 | 34.01 | 8.25 | 7.74 |
| GO4ALIGN [48] | 72.63 | 93.03 | 0.0164 | 27.58 | 10.75 | 8.13 |
| FAIRGRAD [4] | 75.72 | 93.68 | 0.0134 | 32.25 | 3.25 | 5.18 |
| NTKMTL | 73.71 | 92.71 | 0.0136 | 27.21 | 8.50 | 1.92 |
| NTKMTL-SR | 72.58 | 92.93 | 0.0124 | 31.65 | 8.00 | 3.84 |

Table 8: Results on CityScapes (2 tasks) and CelebA (40 tasks) datasets. Each experiment is repeated over 3 random seeds and the mean and stderr are reported.

| | CityScapes | | | | | CelebA |
|---|---|---|---|---|---|---|
| Method | Segmentation | | Depth | | Δm% ↓ | Δm% ↓ |
| | mIoU ↑ | Pix Acc ↑ | Abs Err ↓ | Rel Err ↓ | | |
| NTKMTL (mean) | 73.71 | 92.71 | 0.0136 | 27.21 | 1.92 | -0.77 |
| NTKMTL (stderr) | ±0.17 | ±0.15 | ±0.0005 | ±0.23 | ±0.30 | ±0.37 |
| NTKMTL-SR (mean) | 72.58 | 92.93 | 0.0124 | 31.65 | 3.84 | 0.23 |
| NTKMTL-SR (stderr) | ±0.32 | ±0.23 | ±0.0004 | ±0.35 | ±0.37 | ±0.46 |

Table 9: Results on NYU-v2 dataset (3 tasks). Each experiment is repeated over 3 random seeds and the mean and stderr are reported.

| | Segmentation | | Depth | | Surface Normal | | | | | Δm% ↓ |
|---|---|---|---|---|---|---|---|---|---|---|
| Method | | | | | Angle Dist ↓ | | Within $t°$ ↑ | | | |
| | mIoU ↑ | Pix Acc ↑ | Abs Err ↓ | Rel Err ↓ | Mean | Median | 11.25 | 22.5 | 30 | |
| NTKMTL (mean) | 39.68 | 65.43 | 0.5296 | 0.2168 | 24.24 | 18.63 | 30.74 | 58.72 | 70.78 | -6.99 |
| NTKMTL (stderr) | ±0.51 | ±0.24 | ±0.0008 | ±0.0014 | ±0.07 | ±0.09 | ±0.14 | ±0.19 | ±0.19 | ±0.38 |
| NTKMTL-SR (mean) | 40.23 | 65.28 | 0.5261 | 0.2136 | 24.88 | 19.58 | 29.53 | 56.67 | 69.08 | -5.35 |
| NTKMTL-SR (stderr) | ±0.42 | ±0.26 | ±0.0013 | ±0.0014 | ±0.11 | ±0.13 | ±0.12 | ±0.15 | ±0.16 | ±0.31 |

for NTKMTL-SR on NYUv2 is set to 2. Our setup for the CelebA benchmark aligns with the configuration detailed in [31]. We employ a 9-layer CNN as the network backbone, coupled with separate linear layers for each task. The method is trained for 15 epochs; optimization is carried out using Adam with a batch size of 256.

## C.3 Visualization experiments for the NTK eigenvalues during training

To support the proposed theory, we conducted experiments on NYUv2, visualizing the change in the maximum eigenvalue of the NTK matrix for the three tasks during training under linear scalarization (i.e., equal weighting). On the NYUv2 dataset, the difficulty levels of the three tasks show significant variation. Previous methods generally outperform the Single Task Learning (STL) baseline in segmentation and depth estimation tasks, but almost all of them consistently underperform STL on the surface normal prediction task, leading to a significant task imbalance in the overall results.

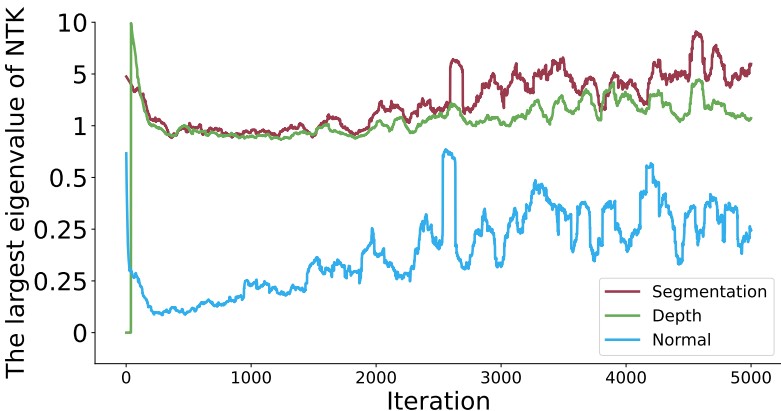

Figure 2: Visualization for the NTK eigenvalues during training.

As shown in Fig. 2, throughout training, the largest eigenvalue corresponding to surface normal prediction remains substantially smaller than those of segmentation and depth estimation. Given that the maximum NTK eigenvalue reflects a task's convergence speed, this observation aligns with the empirical finding that many existing MTL algorithms struggle to converge on surface normal prediction. Although some prior methods (e.g., MGDA) attempt to prioritize the most difficult tasks, their performance in surface normal prediction tasks remains unsatisfactory due to the challenge in accurately quantifying the "difficulty" and "convergence speed" of different tasks. In contrast, by leveraging NTK theory to accurately characterize and balance the convergence speed of each task during training, NTKMTL delivers SOTA results on surface normal prediction and is one of only two methods that outperform single-task learning on all three tasks.

### C.4 Ablation Study on the Hyperparameter $n$

For NTKMTL-SR, the computational cost of calculating the NTK is minimal, allowing us to further investigate the impact of varying mini-batch sizes $n$ on the results. Therefore, we set $n$ to $[1, 2, 3, 4, 6]$ and conduct an ablation study on the QM9 (11-task) benchmark. For each value of $n$, we conduct 3 repeated experiments with different random seeds and calculate the mean and variance for $\Delta m\%$. The results are shown in Fig. 3. When $n = 1$, the performance of NTKMTL-SR is comparable to that of NTKMTL. However, when increasing $n$ from 1 to 2 or more, NTKMTL-SR shows a noticeable improvement in performance, accompanied by a reduction in the variance of performance across repeated experiments. We attribute this to the fact that increasing the number of mini-batches leads to a larger NTK matrix dimension, which in turn reduces stochastic error and allows our method to more accurately characterize the convergence speed of the tasks.

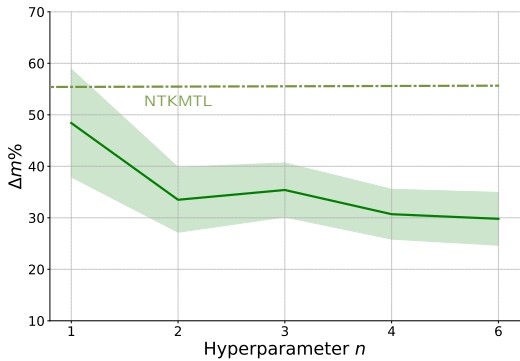

Figure 3: Ablation study on hyperparameter $n$ on QM9. Each experiment is repeated over 3 random seeds, and the mean and stderr are reported.

However, we also find that the results for $n = 4$ and $n = 6$ are almost identical, and the performance differences observed were potentially weaker than the inherent variability stemming from different random seeds. Concurrently, Fig. 4 visualizes the training time per epoch for various $n$ values on QM9. Despite only requiring the computation of the maximum eigenvalue of the NTK matrix with respect to $z$, training a single epoch when $n = 6$ already approached 1.7 times the duration of the LS method. Overall, we posit that the selection of hyperparameter $n$ is a trade-off between performance and training speed, and $n = 4$ generally presents a favorable compromise.

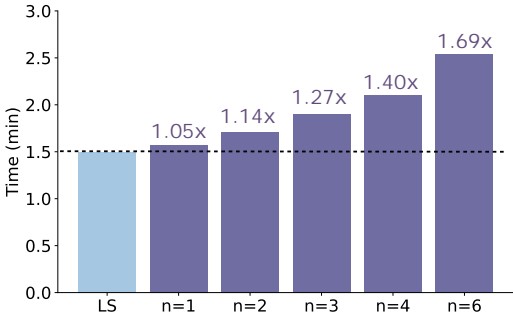

Figure 4: Training time per epoch for LS and NTKMTL-SR (with different hyperparameter $n$) on QM9.

## D    Impact Statement

This work aims to employ NTK theory to investigate the training dynamics of multi-task learning and address the task imbalance issue. We acknowledge the potential implications of our findings on fairness in machine learning systems. Our proposed methods aim to reduce task imbalance and enhance performance equity across different tasks, thereby mitigating biases that may arise in multi-task learning frameworks. We are committed to transparency and responsible dissemination of our results, and we encourage further exploration of the ethical implications of our methodologies.

