# OpenReview forum: "NTKMTL: Mitigating Task Imbalance in Multi-Task Learning from Neural Tangent Kernel Perspective"
_NeurIPS.cc/2025/Conference — NeurIPS 2025 poster_

### Official Review · Reviewer_k73J · 2025-06-11

**Clarity:** 3
**Significance:** 2
**Originality:** 3
**Rating:** 4
**Confidence:** 3

**Summary:**

This paper addresses the persistent challenge of task imbalance in multi-task learning (MTL) by proposing a novel method grounded in Neural Tangent Kernel (NTK) theory. The authors introduce an extended NTK framework to model MTL training dynamics and leverage spectral analysis to derive task-specific weighting. This culminates in two methods: NTKMTL, which explicitly balances task convergence via NTK eigenvalue normalization, and NTKMTL-SR, a computationally efficient approximation using shared representations. Extensive experiments across supervised and reinforcement learning tasks validate the proposed methods, demonstrating competitive or superior performance compared to strong baselines.

**Questions:**

1. Aside from Table 1, most other tables do not report the results of all baseline methods, which makes it difficult to objectively assess the general effectiveness of the proposed method across different scenarios.
2. Are there more ablation studies on hyperparameter n for NTKMTL?

**Ethical Concerns:**

["NO or VERY MINOR ethics concerns only"]

**Final Justification:**

The author has resolved most of my main concerns, including those regarding the incompatibility of NTKMTL-SR (Q1), the weight sum in NTKMTL (Q2), and ablation studies on hyperparameter n (Q4).

The results of all baseline methods (Q3) and further analysis for off-diagonal NTK (Q5) are absent, but this will not particularly affect the contribution and effectiveness of this work.

Therefore, I will maintain my score of borderline accept for this paper.

**Limitations:**

1. Limited Analysis of Off-Diagonal NTK Components: The off-diagonal blocks of the extended NTK matrix (which capture inter-task interactions) are not exploited, potentially missing opportunities to guide task grouping or influence.

**Paper Formatting Concerns:**

No paper formatting concerns.

**Quality:**

3

**Strengths And Weaknesses:**

**Strengths:**
1. Theoretical Innovation: The application of NTK theory to MTL and the formulation of an extended NTK matrix for spectral task analysis are original and theoretically grounded.
3. Clear Algorithmic Design: The derivation of task weights from NTK eigenvalues is logically sound and well-motivated.
2. Broad Experimental Evaluation: The paper evaluates on diverse benchmarks (NYUv2, CityScapes, CelebA, QM9, MT10), with both supervised and reinforcement learning settings, providing strong empirical support.

**Weaknesses:**
1. Incompatibility of NTKMTL-SR with Non-Task-Specific Architectures: The NTKMTL-SR method relies on the presence of task-specific parameters to compute gradients with respect to shared representations. This limits its applicability to architectures where all parameters are shared (i.e., no task-specific heads), making it incompatible with reinforcement learning settings (e.g., MT10) or even large language model scenarios.
2. Unfair Comparison with Other MTL Methods: The weight computation in Eq. 15 does not strictly ensure that the task weights sum to k (the number of tasks). Without a formal guarantee, the sum of weights is generally ≥ k, potentially leading to unfair comparisons with other MTL methods that explicitly constrain weight normalization. **I believe it is critically important to include a fair comparison based on properly normalized weights; otherwise, the strong empirical results presented in the paper lose much of their significance.**

---

> ### Author Rebuttal · Authors · 2025-07-31
>
> We appreciate your recognition of our method's theoretical innovation, clear algorithmic design, and broad experimental evaluation. Thank you for your careful and valuable comments! We hope to address your concerns by answering your questions below.
>
> **Q1: Incompatibility of NTKMTL-SR with Non-Task-Specific Architectures.**
>
> **A1:** We acknowledge that NTKMTL-SR is an approximate algorithm specifically designed for models with task-specific layers, which might limit its direct applicability to settings like MT10 or large language model (LLM) training. However, NTKMTL  supports non-task-specific architectures and has demonstrated superior performance in multi-task reinforcement learning. Additionally, NTKMTL achieves a speed advantage of over 2x compared to some gradient-oriented MTL methods like PCGrad and Nash-MTL, which require additional gradient projection or iterative optimization.
>
> Furthermore, our proposed extended NTK theory in MTL still holds potential for generating new algorithms in these scenarios. For instance, a promising direction involves balancing training losses from different data sources (e.g., code, math) during LLM pre-training by computing the NTK matrix with respect to deep-layer parameters for each data source. We welcome any insights you may have on this!
>
> **Q2: The weight in NTKMTL does not strictly ensure that the task weights sum to k (the number of tasks).**
>
> **A2:** Yes, early MTL methods (e.g. DWA [1]) often constrained the sum of weights to k (the number of tasks). However, the vast majority of recent MTL algorithms (e.g., Nash-MTL [2], FAMO [3], FairGrad [4], GO4Align [5]) do not adhere to this constraint. They operate under the premise that the scale information inherent in the calculated weights is beneficial for training, and that the optimal weight scales differ across various training stages. Forcibly normalizing these weights to k would lead to a loss of this crucial information.
>
> Especially for algorithms like Nash-MTL and FairGrad, the task weights are derived by solving an optimization problem. In such cases, forced normalization would invalidate them as solutions to the optimization problem. Therefore, most contemporary MTL algorithms only require that the weights be bounded. Their specific scale is determined by the algorithm's hyperparameters, which are often tuned to find the optimal settings.
>
> [1] Liu S, Johns E, Davison A J. End-to-end multi-task learning with attention[C]//Proceedings of the IEEE/CVF conference on computer vision and pattern recognition. 2019: 1871-1880.
>
> [2] Navon A, Shamsian A, Achituve I, et al. Multi-task learning as a bargaining game[J]. arXiv preprint arXiv:2202.01017, 2022.
>
> [3] Liu B, Feng Y, Stone P, et al. Famo: Fast adaptive multitask optimization[J]. Advances in Neural Information Processing Systems, 2023, 36: 57226-57243.
>
> [4] Ban H, Ji K. Fair resource allocation in multi-task learning[J]. International Conference on Machine Learning, 2024.
>
> [5] Shen J, Wang Q, Xiao Z, et al. Go4align: Group optimization for multi-task alignment[J]. Advances in Neural Information Processing Systems, 2024.
>
> **Q3: Aside from Table 1, most other tables do not report the results of all baseline methods.**
>
> **A3:** NYUv2, featured in Table 1, is one of the most widely used benchmarks in the MTL field, so nearly all MTL methods conduct experiments on it. Except for QM9, where we re-implemented almost all methods for a fair comparison, results for other MTL methods on other benchmarks were obtained from their original papers.
>
> Since some works have conducted experiments on only two or three benchmarks, we cannot obtain the results from their papers. Many of them overlook CelebA (40-task), a benchmark crucial for evaluating MTL performance with a large number of tasks. This explains why the number of methods shown in Table 2 is fewer than in Table 1. For the CityScapes benchmark, due to space limitations in the main text, we presented its results alongside CelebA in Table 2, showing only 14 MTL methods. We report the detailed results for this benchmark, encompassing 18 methods, in Appendix Table 7.
>
> Thank you for your suggestion. We will consider to re-implement the missing results for some MTL methods on CelebA and integrate them into the latest version of the manuscript.
>
>
> **Q4: Are there more ablation studies on hyperparameter n for NTKMTL?**
>
> **A4:** For NTKMTL-SR, we have conducted a comprehensive ablation study on n , with the detailed results presented in the supplementary material. At your request, we also conducted an ablation study on the hyperparameter n for NTKMTL. We performed experiments CelebA benchmark, and the results are presented in the table below. We maintain consistency with the settings of other MTL methods discussed in the main text, training for 15 epochs on CelebA. For each value of n, we conduct three repeated experiments and report the mean of $\Delta m \\%$.
>
> | Method                     | $\Delta m \\%\downarrow$    | Time per epoch (min) |
> | -------------------------- | ---------------- | -------------------- |
> | NTKMTL (n=1)               | -0.77  | 15.6                 |
> | NTKMTL (n=2)               | -0.95  | 29.3                 |
> | NTKMTL (n=3)               | -0.89  | 42.6                 |
> | NTKMTL (n=4)               | -1.09  | 55.3                 |
>
>
> The table shows that increasing n does yield modest performance gains; however, the computational overhead grows almost linearly with n, in sharp contrast to the negligible cost incurred by NTKMTL-SR. Because the results with n=1 are already highly competitive, and to ensure a fair comparison with other MTL methods, we recommend setting n=1 as the default for NTKMTL. In this case, it aligns with other gradient-oriented methods, requiring k gradient backpropagations for shared parameters per iteration.
>
> **Q5: Further Analysis for Off-Diagonal NTK Components.**
>
> **A5:** Thank you for showing this point. Our motivation is to balance the convergence speeds of different tasks, and thus we mainly require the maximum eigenvalue of the respective NTK matrix of each task to capture the convergence speed. Therefore, we primarily used the diagonal blocks of $\tilde{K}$ for analysis.
>
> We also recognize that the off-diagonal blocks of $\tilde{K}$, specifically $\tilde{K}_{ij}$, might contain rich information of task similarity. These information can potentially lead to new MTL methods based on task grouping. However, as this represents a technical direction orthogonal to our paper's focus of mitigating task imbalance by balancing convergence speeds, we did not elaborate extensively on it here. Instead, we included it in the limitations section for future work.
>
> &nbsp;
>
> These discussions have further enriched the quality of the manuscript. We will incorporate the ablation studies for both NTKMTL and NTKMTL-SR into the appendix, and simultaneously refine parts of the main text for enhanced clarity. If you have any further questions, please don't hesitate to contact us. Thank you again for your valuable time and effort during the review process!

---

> > ### Comment · Reviewer_k73J · 2025-08-03
> >
> > Thank you for your clarifications.

---

### Official Review · Reviewer_9Qpw · 2025-06-27

**Clarity:** 3
**Significance:** 2
**Originality:** 3
**Rating:** 4
**Confidence:** 2

**Summary:**

This paper analyzes the task dominance phenomenon in multi-task learning based on NTK and proposes a method based on NTK singular value balancing to alleviate the task imbalance issue. Experiments were conducted on multi-task supervised learning and multi-task reinforcement learning to validate the effectiveness of the method. However, the paper still lacks evidence to support many of its analyses, and the effectiveness of the proposed method in specific scenarios needs further discussion.

**Questions:**

- On the CITYSCAPES dataset, the MR metric of NTKMTL is significantly worse than the baseline methods (with an average ranking of 7th). How do different datasets affect the effectiveness of NTKMTL? The scenarios in which NTKMTL may fail need further discussion.
- NTKMTL-SR  is an approximation of NTKMTL, but why does the former perform better than the latter in certain scenarios (e.g., Tables 2 and 3)? This is quite confusing.
- How does the task-relatedness/similarity influence task dominance and NTK singular values?

**Ethical Concerns:**

["NO or VERY MINOR ethics concerns only"]

**Final Justification:**

I'm not an expert in this field. I'm considering the decisions of other experts.

**Limitations:**

yes

**Quality:**

3

**Strengths And Weaknesses:**

**Strengths:**

- This paper introduces a new theoretical perspective by analyzing the task dominance problem in the MTL training process from the NTK viewpoint.
- The paper proposes two multi-task balancing methods, NTKMTL and NTKMTL-SR.
- A significant number of experiments are conducted on multi-task supervised learning and reinforcement learning to validate the effectiveness of the proposed methods.

**Weaknesses:**

- The goal of the paper is to analyze the phenomenon of learning speed/NTK singular value dominance in MTL training; however, no visualization experiments are provided to support these claims.
- In line 34, the authors mention "However, different tasks exhibit vastly different loss scales and heterogeneous ultimate loss minima," but this part lacks supporting argumentation or analysis, especially regarding the heterogeneity of the minima.
- The paper mentions that very few works apply NTK to MTL, but the key challenges or difficulties of extending NTK theory from single-task to multi-task need further elaboration.
- Some works in the MTL field have also discussed the task dominance issue in multi-task model training, which should be addressed and referenced, such as Dynamic Priority[1] and AdaTask[2].
- Figure 1 shows the time for training one epoch. Do all methods train for the same number of epochs in the experiment? If not, comparing the time for the entire training phase would be more meaningful.

[1] Guo, Michelle, et al. "Dynamic task prioritization for multitask learning." Proceedings of the European conference on computer vision (ECCV). 2018.

[2] Yang, Enneng, et al. "Adatask: A task-aware adaptive learning rate approach to multi-task learning." Proceedings of the AAAI conference on artificial intelligence. Vol. 37. No. 9. 2023.

---

> ### Author Rebuttal · Authors · 2025-07-31
>
> We appreciate your recognition of our method's new theoretical perspective and extensive experimental validation. Thank you for your careful and valuable comments! We hope to address your concerns by answering your questions below.
>
> **Q1: Visualization experiments for the NTK eigenvalues during training.**
>
> **A1:** Thanks for your suggestion. We agree that experimental visualization is essential to support the proposed theory. Hence, we conducted experiments on NYUv2, visualizing the change in the maximum eigenvalue of the NTK matrix for the three tasks during training under linear scalarization (i.e., equal weighting). As images cannot be uploaded, we present the results in the table below.
>
> |Task|avg over 1000|avg over 2000|avg over 3000|avg over 4000|avg over 5000|
> |-|-|-|-|-|-|
> |segmentation|2.52|2.35|2.92|3.50|3.76|
> |depth|2.71|2.27|2.41|2.68|2.87|
> |surface normal|0.38|0.30|0.33|0.37|0.39|
>
> 'Avg over xx' denotes the mean of the maximum NTK eigenvalues during the first xx iterations. On the NYUv2 dataset, the difficulty levels of the three tasks show significant variation. Previous methods generally outperform the Single Task Learning (STL) baseline in segmentation and depth estimation tasks, but almost all of them consistently underperform STL on the surface normal prediction task, leading to a significant task imbalance in the overall results.
>
> As evident from the table, throughout training, the largest eigenvalue corresponding to surface normal prediction remains substantially smaller than those of segmentation and depth estimation. Given that the maximum NTK eigenvalue reflects a task's convergence speed, this observation aligns with the empirical finding that many existing MTL algorithms struggle to converge on surface normal prediction. Although some prior methods (e.g., MGDA) attempt to prioritize the most difficult tasks, their performance in surface normal prediction tasks remains unsatisfactory due to the challenge in accurately quantifying the "difficulty" and "convergence speed" of different tasks. In contrast, by leveraging NTK theory to accurately characterize and balance the convergence speed of each task during training, NTKMTL delivers SOTA results on surface normal prediction and is one of only two methods that outperform single-task learning on all three tasks.
>
> **Q2: Question about "different tasks exhibit vastly different loss scales and heterogeneous ultimate loss minima".**
>
> **A2:** In MTL, simple equal weighting fails to balance task losses precisely because different tasks inherently possess distinct natures and may utilize distinct loss calculation methods (e.g., cross-entropy loss for classification vs. L2 loss for regression). This leads to vastly different loss scales and different ultimate loss minima, a fact widely acknowledged by many MTL works (e.g., UW, FairGrad, GO4Align). Directly applying equal weighting often results in the training process being dominated by tasks with larger loss scales.
>
> For instance, on the NYUv2 benchmark, the losses for semantic segmentation and depth prediction tasks are typically an order of magnitude higher than that for surface normal prediction. On the even more heterogeneous QM9 dataset, the loss discrepancies among different tasks can span up to three orders of magnitude.
>
> Thanks for your suggestion. We will rephrase this statement in lines 32-39 of the manuscript to enhance clarity.
>
> **Q3: Further elaboration for the key challenges of extending NTK theory from STL to MTL.**
>
> **A3:** In STL, the NTK expression can be directly derived from the gradient flow. However, MTL explicitly divides the objective function into k components, each associated with a specific task. Since all tasks share parameters, the gradient flow of one task also influences the losses of other tasks, which explicitly increases the complexity of the analysis. In MTL, analyzing only the overall loss using STL's NTK theory (Theorem 3.1) fails to capture fine-grained, per-task training dynamics. To fully understand the differences in convergence speed among tasks, it's essential to extend STL's NTK theory, which poses a new theoretical challenge.
>
> We analyzed the specific scenario of MTL and proposed Theorem 3.2 and Proposition 3.3 (proofs detailed on pages 15-17). The most intuitive physical interpretation these theories provide is that with the introduction of weighting coefficients $w_i$, the new NTK matrix for each task becomes $w_i^2K_{ii}$, and its maximum eigenvalue is accordingly scaled to $w_i^2 \lambda_i$. This ultimately led to our final method design.
>
> Thanks for your suggestion. We will further clarify the necessity of our proposed theory in lines 175-197 of the method section and provide a smoother transition to the method design.
>
> **Q4: Missing References.**
>
> **A4:** Thank you for the additions! We have previously read both of these two papers. Dynamic Task Prioritization (DTP), in particular, is a significant work in the MTL field. It was an oversight on our part to have omitted these two papers from the related work section. We've now included them in the manuscript.
>
> **Q5: Do all methods train for the same number of epochs in the experiment?**
>
> **A5:** Yes, to ensure fairness, all methods are trained for the same number of epochs across all benchmarks. Therefore, the training time per epoch can essentially represent the comparison of total training duration.
>
> **Q6: On CITYSCAPES, the MR of NTKMTL is worse than some baseline methods. How do different datasets affect the effectiveness of NTKMTL?**
>
> **A6:** While our method's performance varies across different benchmarks, a consistent pattern emerges: it consistently optimizes more difficult tasks more effectively, leading to more balanced optimization and achieving superior average performance.
>
> When evaluating MTL methods, $\Delta m\\%$ is typically the most intuitive metric as it relies solely on the objective performance of the MTL method on a specific benchmark. Mean Rank (MR), however, depends on the number and performance of other MTL methods used for comparison, and thus usually serves as a supplementary criterion when $\Delta m\\%$ values are similar.
>
> **Why does NTKMTL exhibit suboptimal MR performance on the CityScapes dataset?** We report the detailed performance of 17 different MTL methods on the CityScapes in Appendix Table 7 (page 19). The CityScapes task comprises two subtasks: segmentation and depth prediction. Due to the disparity in task difficulty, most prior methods excelled at the segmentation task but performed significantly worse than the STL baseline on the depth prediction task, exhibiting severe task imbalance. Our method effectively balances the performance of different tasks, achieving significant progress in the depth task. However, this also implies a slight decrease in performance on the segmentation task, placing our method at a disadvantage in MR calculation.
>
> In summary, NTKMTL's suboptimal MR performance on CityScapes does not indicate a failure of the method on this dataset. In fact, it effectively mitigates task imbalance and achieves the optimal $\Delta m\\%$.
>
> **Q7: Why does NTKMTL-SR perform better than NTKMTL in certain scenarios?**
>
> **A7:** In Table 2, NTKMTL-SR does not outperform NTKMTL in terms of $\Delta m\\%$ (0.23 vs. -0.77), achieving only a slightly better Mean Rank (MR) (4.33 vs. 4.35). As stated in A6, the MR metric is susceptible to changes based on the inclusion or exclusion of other comparative MTL methods.
>
> However, on the QM9 dataset, NTKMTL-SR indeed surpasses NTKMTL and achieves state-of-the-art performance. We also analyzed this phenomenon, attributing it to the relatively small number of model parameters used in this regression task; consequently, even with approximation, the training dynamics of different tasks can be captured effectively. As a result, a larger n produces a more accurate estimate of convergence speed. To further validate this, we conducted detailed ablation experiments in the supplementary material, where we set n to [1, 2, 3, 4, 6] for NTKMTL-SR and performed three experiments for each setting, reporting the mean and error bars. As shown in Fig. 2 of the supplementary material, when n=1, the performance of NTKMTL-SR is comparable to that of NTKMTL. However, when increasing n from 1 to 2 or more, NTKMTL-SR shows a noticeable improvement in performance, accompanied by a reduction in the variance of performance across repeated experiments.
>
> **Q8: How does the task-relatedness/similarity influence task dominance and NTK singular values?**
>
> **A8:** During our experiments, we observed that tasks exhibiting clear similarities (e.g., tasks 7-10 on QM9) not only have similarly scaled maximum eigenvalues of their NTK matrices but also show consistent trends in their evolution throughout training. Consequently, NTKMTL assigns essentially equal weights to these tasks to facilitate balanced optimization. (As shown in Table 3, these highly correlated tasks exhibit very close final MAE values.) We have also visualized this phenomenon, and the images and corresponding analysis will be included in the latest version of the appendix.
>
> Furthermore, we acknowledge that the off-diagonal blocks of $\tilde{K}$, specifically $\tilde{K}_{ij}$, contain rich information. These blocks can, to some extent, represent task similarity and potentially lead to new MTL methods based on task grouping. However, due to the orthogonality of the task grouping technical approach to our current method, we include it in the limitations section for future work.
>
> &nbsp;
>
> These discussions have further enriched the quality of the manuscript. We will integrate all rebuttal content, adding visualizations as figures to the appendix, and simultaneously revise and refine certain statements in the main text. If you have any further questions, please don't hesitate to contact us. Thank you again for your valuable time and effort during the review process!

---

> > ### Author Response · Authors · 2025-08-07
> > **Sincerely looking forward to your reply**
> >
> > Dear Reviewer,
> >
> > We sincerely thank you for your thoughtful and detailed feedback.
> >
> > We have carefully addressed your concerns in our rebuttal, including:
> >
> > - Additional visualization experiments for the NTK eigenvalues during training, accompanied by a corresponding analysis.
> > - Clearer explanations of specific details within the manuscript and the inclusion of previously omitted references.
> > - Detailed analysis of experimental performance, including the analysis of the effectiveness of NTKMTL on different benchmarks and an in-depth analysis of the superior performance of NTKMTL-SR in certain scenarios, validated by ablation studies.
> >
> >
> > As the discussion period is nearing its end, we want to ensure that our responses have satisfactorily addressed your questions. Please do not hesitate to contact us if you have any further concerns.
> >
> > Sincerely,
> >
> > Submission 9180 authors.

---

> > > ### Comment · Reviewer_9Qpw · 2025-08-08
> > >
> > > Thank you for the author's detailed reply. I have no further questions.

---

> > > > ### Author Response · Authors · 2025-08-08
> > > > **Glad to hear your concerns are resolved**
> > > >
> > > > Thank you for your feedback! We are very glad to hear that our responses have successfully addressed your concerns. We believe our discussions have further improved the quality of the manuscript. We would appreciate it if you could update your rating accordingly when submitting the final rate.

---

### Official Review · Reviewer_yMmP · 2025-06-28

**Clarity:** 4
**Significance:** 3
**Originality:** 4
**Rating:** 5
**Confidence:** 4

**Summary:**

This paper introduces NTKMTL, a novel Multi-Task Learning (MTL) method designed to mitigate task imbalance, a prevalent challenge where certain tasks dominate the training process. The core contribution is the application of Neural Tangent Kernel (NTK) theory to analyze and balance the convergence speeds of multiple tasks within an MTL system. The authors propose an extended NTK matrix for MTL, showing that tasks associated with larger NTK eigenvalues converge faster and can dominate others. To address this, NTKMTL assigns weights to tasks based on the maximum eigenvalues of their respective NTK matrices, aiming to normalize their convergence speeds. An efficient variant, NTKMTL-SR, is also introduced, which leverages shared representation approximation to improve computational efficiency, especially for scenarios with many tasks. Extensive experiments across multi-task supervised learning (NYUv2, CityScapes, QM9, CelebA) and multi-task reinforcement learning (MT10) benchmarks demonstrate that NTKMTL and NTKMTL-SR achieve state-of-the-art performance.

**Questions:**

Impact of n on NTK Estimation and Performance: The paper uses different n values for NTKMTL (n=1) and NTKMTL-SR (n=4), noting that larger n yields "a more accurate estimate of convergence speed". Please provide an ablation study on how varying n affects the accuracy of the NTK eigenvalue estimation and, consequently, the overall performance of both NTKMTL and NTKMTL-SR.

**Ethical Concerns:**

["NO or VERY MINOR ethics concerns only"]

**Final Justification:**

I acknowledge that i have read the authors' rebuttal to all reviewers. The response by the authors and the additional clarifications provided further solidifies my initial assessment of the work. I will keep my original high rating of 5.

**Limitations:**

The authors have adequately discussed the limitations of their work. They acknowledge that their current weight design primarily focuses on task-specific NTK matrices ($K_{ii}$) and that further analysis of the full extended NTK matrix ($\tilde{K}$), particularly the off-diagonal ($K_{ij}$) blocks, is an area for future investigation to reveal insights into task interactions. They also discuss the computational efficiency of their methods, contrasting NTKMTL with the more efficient NTKMTL-SR. Furthermore, they explicitly mention the challenge of applying NTKMTL-SR to multi-task reinforcement learning due to the difficulty in partitioning parameters into shared and task-specific components in single policy learning scenarios. Finally, the paper includes an "Impact Statement" addressing potential positive societal impacts regarding fairness and performance equity in machine learning systems, reflecting a commitment to responsible dissemination.

**Paper Formatting Concerns:**

Line 79: "taks" -> "task"
Line 531: "task" -> "tasks"

**Quality:**

4

**Strengths And Weaknesses:**

Strengths:
- Novelty: This paper presents a novel approach to address a significant problem in Multi-Task Learning, namely task imbalance. The application of Neural Tangent Kernel (NTK) theory to MTL is a significant and original contribution, providing a new lens through which to understand and mitigate this issue.
- Strong Theoretical Foundation: The paper rigorously extends the foundational NTK theory to the MTL context, formally defining an extended NTK matrix and deriving its evolution equation. This theoretical framework allows for a clear characterization of convergence speeds based on NTK eigenvalues.
- Effective Task Imbalance Mitigation: NTKMTL's strategy of weighting tasks based on their NTK eigenvalues effectively balances convergence speeds, leading to demonstrably improved overall performance and more balanced optimization across tasks.
State-of-the-Art Performance: The experimental results are comprehensive and show impressive state-of-the-art performance across a diverse range of benchmarks, including computer vision, molecular property prediction, and reinforcement learning.
- Computational Efficiency with NTKMTL-SR: The introduction of NTKMTL-SR is a crucial practical improvement. By leveraging shared representations, it significantly reduces computational cost, making the method feasible for tasks with a large number of components (e.g., 40 tasks on CelebA) while maintaining competitive performance.
- Commitment to Reproducibility: The authors provide detailed algorithmic descriptions, pseudocode, and experimental settings. Their effort to reproduce and re-evaluate baselines on QM9 under optimized settings for a fair comparison is commendable and enhances the credibility of their results. The authors also commit to making their code publicly available with camera-ready submission.

Weaknesses:
- Limited Deep Dive into Extended NTK (K̃): While the paper introduces the extended NTK matrix K̃ and acknowledges its potential for deeper analysis (e.g., off-diagonal Kij blocks for task interactions), the primary weight design focuses only on the task-specific diagonal blocks Kii. This leaves a significant portion of the newly introduced theoretical framework underexplored in the current method. For a theoretically driven paper, this feels like an incomplete utilization of the proposed extended kernel.

---

> ### Author Rebuttal · Authors · 2025-07-31
>
> We appreciate your recognition of our method's novelty, strong theoretical foundation, effectiveness, and efficiency. Thank you for your careful and valuable comments! We hope to address your concerns by answering your questions below.
>
> **Q1: Ablation Study on Hyperparameter n**
>
> **A1:** We also noticed that exploring the impact of n as a hyperparameter on experimental results is meaningful, especially for NTKMTL-SR. Therefore, we conducted a comprehensive ablation study on n for NTKMTL-SR, with the detailed results presented in the supplementary material. We set n to $[1, 2, 3, 4, 6]$ and conduct an ablation study on the QM9 (11-task) benchmark. For each value of n, we perform 3 repeated experiments with different random seeds and calculate the mean and error bar for $\Delta m\\%$. Detailed results and analysis are available in the supplementary material, and we have also excerpted them here for convenience.
>
> | n | $\Delta m\\%\downarrow$ | time per epoch (min) |
> |---|---|---|
> | 1 | 48.4 | 1.57 |
> | 2 | 33.5 | 1.71 |
> | 3 | 35.4 | 1.90 |
> | 4 | 30.7 | 2.10 |
> | 6 | 29.8 | 2.54 |
>
> The results indicate that increasing n from 1 to 2 or more shows a noticeable improvement in NTKMTL-SR's performance, accompanied by a reduction in the error bar of performance across repeated experiments. We attribute this to the fact that increasing the number of mini-batches leads to a larger NTK matrix dimension, which in turn reduces stochastic error and allows our method to more accurately characterize the convergence speed of the tasks. However, since further increasing n (i.e., from 4 to 6) does not yield significant additional benefits and introduces further computational overhead, we posit that the selection of hyperparameter n is a trade-off between performance and training speed, and n=4 generally presents a favorable compromise.
>
> Additionally, at your request, we also conducted an ablation study on the hyperparameter n for NTKMTL. Due to limited rebuttal time, these experiments were performed on the smaller CelebA benchmark, and the results are presented in the table below. We maintain consistency with the settings of other MTL methods discussed in the main text, training for 15 epochs on CelebA. For each value of n, we conduct three repeated experiments and report the mean of $\Delta m \\%$.
>
> | Method                     | $\Delta m \\%\downarrow$    | Time per epoch (min) |
> | -------------------------- | ---------------- | -------------------- |
> | NTKMTL (n=1)               | -0.77  | 15.6                 |
> | NTKMTL (n=2)               | -0.95  | 29.3                 |
> | NTKMTL (n=3)               | -0.89  | 42.6                 |
> | NTKMTL (n=4)               | -1.09  | 55.3                 |
>
>
> The table shows that increasing n does yield modest performance gains; however, the computational overhead grows almost linearly with n, in sharp contrast to the negligible cost incurred by NTKMTL-SR. Because the results with n=1 are already highly competitive, and to ensure a fair comparison with other MTL methods, we recommend setting n=1 as the default for NTKMTL. In this case, it aligns with other gradient-oriented methods, requiring k gradient backpropagations for shared parameters per iteration.
>
> **Q2: Further analysis of the full extended NTK matrix $\tilde{K}$, particularly the off-diagonal $\tilde{K}_{ij}$ blocks.**
>
> **A2:** Thank you for showing this point. Our motivation is to balance the convergence speeds of different tasks, and thus we mainly require the maximum eigenvalue of the respective NTK matrix of each task to capture the convergence speed. Therefore, we primarily used the diagonal blocks of $\tilde{K}$ for analysis.
>
> We also recognize that the off-diagonal blocks of $\tilde{K}$, specifically $\tilde{K}_{ij}$, might contain rich information about task similarity. These information can potentially lead to new MTL methods based on task grouping. However, as this represents a technical direction orthogonal to our paper's focus of mitigating task imbalance by balancing convergence speeds, we did not elaborate extensively on it here. Instead, we included it in the limitations section for future work.
>
> **Q3: About typographical errors.**
>
> **A3:** Thank you for your careful review! We've corrected these typos, and they will be eliminated in the latest version of the paper.
>
> &nbsp;
>
> These discussions have further enriched the content of this paper. We will integrate the relevant ablation studies into the main text and Appendix, while also carefully reviewing the manuscript to eliminate any typos. Thank you again for your high appreciation of our work and for the valuable time and effort during the review process!

---

### Official Review · Reviewer_Gi8p · 2025-07-02

**Clarity:** 3
**Significance:** 2
**Originality:** 3
**Rating:** 4
**Confidence:** 3

**Summary:**

This paper addresses the common issue of task imbalance in Multi-Task Learning (MTL) and proposes a novel optimization framework called NTKMTL based on Neural Tangent Kernel (NTK) theory. The method analyzes the training dynamics of multiple tasks by extending the NTK matrix, utilizes spectral analysis to evaluate the convergence speed of each task, and designs a task weight allocation strategy based on the maximum eigenvalue of the NTK to balance the training speeds of different tasks, thereby alleviating task imbalance. Furthermore, the authors introduce an approximate variant, NTKMTL-SR, which leverages Shared Representation to reduce computational overhead. Experiments were conducted on multi-task supervised learning, multi-task regression, and multi-task reinforcement learning across several mainstream datasets, achieving better performance.

**Questions:**

1. What are the reasons why the proposed method achieves better results compared to other approaches?
2. Are there any modules or parameter choices in the proposed method that are particularly critical?
3. What are the reasons for the varying performance of the proposed method across different tasks?

**Ethical Concerns:**

["NO or VERY MINOR ethics concerns only"]

**Final Justification:**

Most of my concerns have been resolved, and I will maintain my rating.

**Limitations:**

Yes

**Quality:**

2

**Strengths And Weaknesses:**

Strengths:
1. The method is reasonable, and the task weighting strategy based on the maximum NTK eigenvalue has a clear meaning. The weights are inversely proportional to the convergence speed of the tasks, effectively adjusting the learning speed between tasks.
2. The experiments are comprehensive and diverse, covering multiple supervised and reinforcement learning tasks such as NYUv2, Cityscapes, CelebA, QM9, and MT10. The evaluations are thorough, showcasing the broad adaptability of the proposed method.

Weaknesses:
1. This paper lacks more in-depth analysis and comparison. It mainly describes the rationale of the proposed method and the final accuracy achieved. In practice, balancing tasks based on gradients is a common approach, and other works also have their own validity. When compared with other studies, the paper lacks deeper theoretical or experimental analysis to explain why the proposed method is superior.
2. This paper lacks ablation studies. Although extensive experiments were conducted on various datasets, the results only compare the overall performance of the proposed method. It lacks ablation studies to validate the contributions of specific modules and the choices of certain parameters.
3. Although this paper shows improvement on certain datasets, the improvement is not particularly significant, and the method does not achieve the best performance on many subtasks. A deeper analysis of the underlying reasons behind these experimental results would make the work more robust and comprehensive.

---

> ### Author Rebuttal · Authors · 2025-07-31
>
> We appreciate your recognition of our method's rationality, the comprehensiveness and diversity of our experiments, and the broad adaptability of the proposed method. Thank you for your careful and valuable comments! We hope to address your concerns by answering your questions below.
>
> **Q1: Why does the proposed method achieve better results compared to other approaches (e.g. other methods that balance tasks based on gradients)?**
>
> **A1:** The superior performance of the proposed NTKMTL can be attributed to its ability to effectively balance the convergence rates of individual tasks, which fosters more equitable optimization and ultimately yields enhanced overall performance.
>
> **Why balanced optimization counts:** Recent works (e.g., FairGrad[1] and GO4Align[2]) have demonstrated that more balanced cross-task performance typically yields better overall results (See Fig.2 in [2]).  To more clearly elucidate the reason for NTKMTL's superior performance, we analyze it using the NYUv2 benchmark as an example. On the NYUv2 dataset, the difficulty levels of the three tasks show significant variation. Previous methods generally outperform the Single Task Learning (STL) baseline in the tasks of semantic segmentation and depth estimation, but almost all of them consistently underperform STL on the surface normal prediction task, leading to a significant task imbalance in the overall results. Although some prior methods (e.g., MGDA) attempt to prioritize the most difficult tasks, their performance in surface normal prediction tasks remains unsatisfactory due to the challenge in accurately quantifying the "difficulty" and "convergence speed" of different tasks. In contrast, by leveraging NTK theory to accurately characterize and balance the convergence speed of each task during training, NTKMTL demonstrates strong performance in the surface normal prediction task. Notably, among all existing methods, only NTKMTL and GO4Align consistently outperform the STL baseline across all three tasks, achieving a more balanced optimization.
>
> [1] Ban H, Ji K. Fair resource allocation in multi-task learning[J]. International Conference on Machine Learning, 2024.
>
> [2] Shen J, Wang Q, Xiao Z, et al. Go4align: Group optimization for multi-task alignment[J]. Advances in Neural Information Processing Systems, 2024.
>
>
> **Experimental validation:** To further substantiate this, we conducted additional experiments and visualized: **1)** the maximum eigenvalue of the NTK matrix for different tasks during NYUv2 training, and **2)** the relationship between cross-task performance variance and $\Delta m\\%$ for various methods on NYUv2. As images cannot be uploaded, we present these findings in the tables below.
>
> **1). The maximum eigenvalue of the NTK matrix for different tasks during NYUv2 training.**
>
> | Task    | avg over 1000 | avg over 2000 | avg over 3000 | avg over 4000 | avg over 5000 |
> |--|-|--|-|--|-|
> | segmentation          | 2.52   | 2.35          | 2.92          | 3.50          | 3.76          |
> | depth prediction      | 2.71          | 2.27          | 2.41          | 2.68          | 2.87          |
> | surface normal prediction | 0.38      | 0.30          | 0.33          | 0.37          | 0.39          |
>
> In the table above, we report the evolution of the largest eigenvalue of the NTK matrix for each task when utilizing conventional linear scalarization method. ‘Avg over xxxx’ denotes the mean of the maximum NTK eigenvalues during the first xxxx iterations. Throughout training, the largest eigenvalue corresponding to surface-normal prediction remains substantially smaller than those of segmentation and depth estimation. Since the maximum NTK eigenvalue reflects the convergence speed of a task, this observation aligns with the empirical finding that many existing MTL algorithms struggle to converge on surface-normal prediction. By accurately quantifying these differing convergence rates and re-allocating weights accordingly, NTKMTL attains more balanced performance, delivers state-of-the-art results on surface-normal prediction, and is one of only two methods that outperform single-task learning on all three tasks.
>
> **2). The relationship between cross-task performance variance and $\Delta m\\%$ on NYUv2.**
>
> | Method   | $\Delta m\\%\downarrow$ | Var[$\Delta m\\%$] |
> | -------- | -| -- |
> | LS       | 5.59         | 259.13            |
> | SI       | 4.39         | 247.77            |
> | UW       | 4.05         | 190.73            |
> | PCGrad   | 3.97         | 173.66            |
> | DWA      | 3.57         | 191.93            |
> | CAGRAD   | 0.20         | 137.94            |
> | Nash-MTL | -4.04        | 108.03            |
> | FAMO     | -4.10        | 74.66             |
> | FairGrad | -4.66        | 71.59             |
> | GO4Align | -6.08        | 60.78             |
> | NTKMTL   | -6.99        | 62.85             |
>
> The table above reports $\Delta m\\%$ (average performance drop) and Var[$\Delta m\\%$] (variance of the performance drop across tasks) on NYUv2 for NTKMTL and other 10 existing MTL methods. Notably, the performance $\Delta m\\%$ of different MTL methods exhibits a striking positive correlation with cross-task performance variance, and methods with smaller performance variance tend to achieve better overall performance. This is precisely why “mitigating task imbalance” is of paramount importance in the field of multi-task learning, and it also supports why NTKMTL can achieve a superior $\Delta m\\%$ through more balanced optimization.
>
>
> **Q2: Are there any modules or parameter choices in the proposed method that are particularly critical (Have any ablation studies been conducted)?**
>
> **A2:** Since the weights are derived by computing the maximum eigenvalue of the NTK matrix for different tasks, the samples within a batch need to be split into n mini-batches.
> While a larger n can lead to a more precise NTK representation, it also incurs greater computational overhead.
>
> We also noticed that exploring the impact of n as a hyperparameter on experimental results is meaningful, especially for NTKMTL-SR. Therefore, we conducted a comprehensive ablation study on n for NTKMTL-SR, with the detailed results presented in the supplementary material. We set n to $[1, 2, 3, 4, 6]$ and conduct an ablation study on the QM9 (11-task) benchmark. For each value of n, we perform 3 repeated experiments with different random seeds and calculate the mean and error bar for $\Delta m\\%$. Detailed results and analysis are available in the supplementary material, and we have also excerpted them here for convenience.
>
> | n | $\Delta m\\%\downarrow$ | time per epoch (min) |
> |---|---|---|
> | 1 | 48.4 | 1.57 |
> | 2 | 33.5 | 1.71 |
> | 3 | 35.4 | 1.90 |
> | 4 | 30.7 | 2.10 |
> | 6 | 29.8 | 2.54 |
>
> The results indicate that increasing n from 1 to 2 or more shows a noticeable improvement in NTKMTL-SR's performance, accompanied by a reduction in the error bar of performance across repeated experiments. We attribute this to the fact that increasing the number of mini-batches leads to a larger NTK matrix dimension, which in turn reduces stochastic error and allows our method to more accurately characterize the convergence speed of the tasks. However, since further increasing n (i.e., from 4 to 6) does not yield significant additional benefits and introduces further computational overhead, we posit that the selection of hyperparameter n is a trade-off between performance and training speed, and n=4 generally presents a favorable compromise.
>
> For NTKMTL, due to the memory and time cost associated with fully computing the gradients of n mini-batches for shared parameters to construct the NTK matrix, we set n=1 to ensure fairness when comparing with other methods. In this case, it aligns with other gradient-oriented methods, requiring k gradient backpropagations for shared parameters per iteration. Additionally, during the rebuttal period, we conducted further ablation experiments on the hyperparameter n for NTKMTL. You can refer to A1 in our response to Reviewer 2 (yMmP) for details.
>
>
> **Q3: Why does the proposed method exhibit varying performance across different tasks?**
>
> **A3:** In the field of MTL, achieving optimal performance across all subtasks on a given benchmark simultaneously is challenging, as it essentially means directly extending the Pareto frontier. **While our method's performance varies across different benchmarks, a consistent pattern emerges: it consistently optimizes more difficult tasks more effectively, leading to more balanced optimization, effectively mitigating task imbalance, and achieving superior average performance.**
>
> For instance, on NYUv2, NTKMTL achieved  SOTA results on the most challenging surface normal prediction task. It also obtained the optimal MR and $\Delta m\\%$. On the CityScapes benchmark, most prior methods excelled at the segmentation task but performed significantly worse than the Single-Task Learning (STL) baseline on depth prediction. NTKMTL, however, better balanced the performance of these two tasks and achieved the optimal $\Delta m\\%$.
>
> &nbsp;
>
> These discussions have further enriched the content of this paper. We will integrate the rebuttal content and update the manuscript accordingly. Specifically, the visualizations from A1 regarding the maximum eigenvalue of the NTK matrix for different tasks during training and the relationship between cross-task performance variance and $\Delta m\\%$ on various benchmarks will be added as figures to the Appendix. The corresponding analysis from A1 will also be incorporated into the main experimental section. The results of the ablation studies will be integrated into both the main text and the Appendix. The discussion from A3 will be added around lines 252-270 in Section 4.2 to enhance the analysis of the experimental results.
>
> If you have any further questions, please do not hesitate to contact us. Thank you again for your valuable time and effort during the review process!

---

> > ### Author Response · Authors · 2025-08-07
> > **Sincerely looking forward to your reply**
> >
> > Dear Reviewer,
> >
> > We sincerely thank you for your thoughtful and detailed feedback.
> >
> > We have carefully addressed your concerns in our rebuttal, including:
> >
> > - Detailed explanation of NTKMTL's capacity to achieve superior performance, which is further supported by two additional visualization results.
> > - Thorough ablation studies of the hyperparameter n in our methods, along with a corresponding analysis.
> > - Analysis of NTKMTL's performance across different benchmarks, detailing the consistent patterns observed in its behavior.
> >
> > As the discussion period is nearing its end, we want to ensure that our responses have satisfactorily addressed your questions. Please do not hesitate to contact us if you have any further concerns.
> >
> > Sincerely,
> >
> > Submission 9180 authors.

---

### Decision · Program_Chairs · 2025-09-17

**Decision:**

Accept (poster)

**Comment:**

Thank you for your submission to the NeurIPS 2025. This paper analyzes the training dynamics in MTL by leveraging Neural Tangent Kernel (NTK) theory and propose a new MTL method, NTKMTL. The method is reasonable, equied with theoretical guarantee and comprehensive experiments. Although everyone has pointed out some merits, they also have raised some concerns, such as missing in-depth analysis. During the rebuttal period, the authors’ feedback has helped on clarifying the reviewers’ concerns. Thus, the average score 4.25 is both above the average levels. Based on the current reviews and closed-door reviewer discussions, every reviewer seems to be fine with an acceptance.